# Search-Augmented Masked Diffusion Models for Constrained Generation

## Abstract

Discrete diffusion models generate sequences by iteratively denoising samples corrupted by categorical noise, offering an appealing alternative to autoregressive decoding for structured and symbolic generation. However, standard training targets a likelihood-based objective that primarily matches the data distribution and provides no native mechanism for enforcing hard constraints or optimizing non-differentiable properties at inference time. This work addresses this limitation and introduces *SearchDiff*, a training-free neurosymbolic inference framework that integrates informed search directly into the reverse denoising process. At each denoising step, the model predictions define a proposal set that is optimized under a user-specified property satisfaction, yielding a modified reverse transition that steers sampling toward probable and feasible solutions. Experiments in biological design and symbolic reasoning illustrate that *SearchDiff* substantially improves constraint satisfaction and property adherence, while consistently outperforming discrete diffusion and autoregressive baselines.

## 1. Introduction

Discrete diffusion models have emerged as a powerful paradigm for generative modeling of discrete sequences and attracted significant interest due to their ability to model complex dependencies and global structure (Sahoo et al., 2024; Austin et al., 2023). Unlike autoregressive models, which are constrained by a fixed generation order, discrete diffusion models treat sequence generation as a global denoising process, enabling joint modeling of long-range dependencies and global structural patterns. This property makes discrete models especially attractive for scientific and symbolic domains, in which properties (e.g., drug-likeness,

biological activity) and constraints (e.g., chemical validity, logical consistency) often depend on the entire sequence structure and can be assessed only when this global view is available (Lee et al., 2025; Schiff et al., 2024; Cardei et al., 2025). However, while constraints are key in these domains, standard discrete diffusion models are trained to optimize a likelihood-based objective that encourages matching the training data distribution, but does not provide a native mechanism for enforcing hard constraints or optimizing non-differentiable properties at inference time.

To mitigate this issue, recent work on controllable discrete diffusion incorporates property guidance or into the generation process. Nevertheless, most methods still require additional training or fine-tuning, or else rely on approximations that have been shown to introduce bias and degrade generation quality (Schiff et al., 2024; Cardei et al., 2025; Nisonoff et al., 2024). Moreover, fine-tuning a generative model for each new constraint fundamentally alters the original model and is impractical in settings where constraints vary across tasks or application contexts. These limitations highlight the *need for a training-free, inference-time mechanism that can enforce a diverse body of constraints while preserving the generative strengths of discrete diffusion models*.

This work addresses this need and introduces a novel integration of neurosymbolic search into discrete diffusion models. The proposed method, *Search-Augmented Masked Diffusion* (SearchDiff), builds on the insight that the parallel denoising dynamics of masked diffusion expose a natural interface for sequence-level refinement. SearchDiff executes these refinements by embedding a local search step into each reverse denoising iteration, using the model's predicted distribution to first generate high-quality proposals, and then to navigate the discrete space toward feasible regions. This integration enables several key properties: first, it naturally exploits the non-determinism of parallel denoising with that of classical non-deterministic search methods, enabling global sequence-level refinements; second, it enables the use of non-differentiable constraints, allowing it to accommodate complex functions such as chemical simulators or symbolic solvers; third, being training-free, it directly leverages the model's learned inductive biases during the proposal and search process; and finally, by enforcing domain constraints at inference time, it allows the model to sample solutions that are feasible, but low-support under the

[1]Anonymous Institution, Anonymous City, Anonymous Region, Anonymous Country. Correspondence to: Anonymous Author <anon.email@domain.com>.

Preliminary work. Under review by the International Conference on Machine Learning (ICML). Do not distribute.

*Figure 1.* Illustration of the *SearchDiff* method. **Task**: QED-optimized molecular generation.

base model. Figure 1 illustrates an example of the procedure, and shows how SearchDiff steers the denoising trajectory towards maximizing a drug-likeness score (QED) through iterative structural refinements.

**Contributions.** This paper makes three key contributions:

1. It introduces *SearchDiff*, a training-free inference method that embeds informed discrete search into each reverse denoising step of discrete diffusion to enforce sequence-level constraints.

2. It formalizes constrained discrete diffusion via constraint-violation functions and defines search-guided reverse transitions that incorporate constraint and property feedback while preserving the model's inductive biases. This enables non-differentiable constraints (e.g., simulators or symbolic solvers) to be enforced at inference time without model modification.

3. Finally, it evaluates the method on five tasks spanning biological sequence generation (molecules, peptides, and transfer RNA) and constrained symbolic reasoning (Sudoku and Boolean satisfiability), showing large and consistent gains in both constraint satisfaction and property adherence over prior methods. For molecular generation, SearchDiff achieves up to a fourfold increase in the fraction of samples satisfying Synthetic Accessibility relative to standard diffusion while also attaining the highest average drug-likeness; for Boolean SAT, it raises accuracy from $9.6\%$ to $76.0\%$ (nearly $8\times$) and outperforms strong diffusion and autoregressive baselines.

## 2. Related work

The need for controllable generation in discrete diffusion has led to the development of guidance methods to bias the learned distribution towards desired properties. For example, in the context of protein design, Gruver et al. (2023) introduce gradient-guided sampling by updating the denoiser's continuous hidden states and enabling gradient-

based steering despite the non-differentiability of discrete tokens. Schiff et al. (2024) later provided a principled derivation of classifier-free and classifier-based guidance for discrete diffusion. Projected diffusion methods (Cardei et al., 2025; Christopher et al., 2025) instead enforce feasibility by casting each denoising step as an optimization problem that projects the predicted categorical distribution onto a predefined feasible set. However, these approaches typically rely on smooth gradient signals and/or additional training.

Because SearchDiff performs search at inference time, it is also related to search-based controllable decoding for autoregressive models. In the autoregressive setting, controllable decoding may also be approached as a search problem; for example, lexically constrained beam search enforces required words or phrases by tracking constraint coverage during expansion, as in Grid Beam Search (Hokamp & Liu, 2017) and Dynamic Beam Allocation (Post & Vilar, 2018). For richer structure, NeuroLogic Decoding injects predicate-logic lexical constraints into beam search (Lu et al., 2021b), and subsequent work adds lookahead heuristics to better satisfy logical and structural requirements (Lu et al., 2021a). Related approaches also reweight next-token scores using learned predictors to steer generation toward target attributes without modifying the base model (Yang & Klein, 2021). However, because these methods commit to a fixed left-to-right order, their search operates on partial sequences and must predict global satisfaction from incomplete prefixes, which can cause sequential error accumulation: early token choices may irreversibly steer decoding into regions where the final constraint is difficult, or impossible, to meet. Masked diffusion avoids this prefix limitation because each step operates on a specified full sequence.

In contrast, SearchDiff leverages the parallel denoising dynamics of masked diffusion to support global sequence refinement throughout the reverse process. The integration of discrete search within each denoising step is key to align sampling with user-defined constraints using black-box and non-differentiable constraint-violation functions, while preserving the diffusion model's learned proposal distribution and requiring neither additional training nor gradient access.

## 3. Preliminaries

Consider a sequence $\boldsymbol{x} = (x^1, \ldots, x^L)$ of $L$ discrete tokens, each taking value from a vocabulary $\mathcal{V}$. Discrete diffusion models define a Markov noising process that progressively corrupts an initial sequence $\boldsymbol{x}_0 = \boldsymbol{x}$ into a simple prior distribution $\boldsymbol{x}_T$. This time-indexed transformation $\boldsymbol{x}_0 \to \boldsymbol{x}_T$ is called the *forward process* and is parameterized by a set of transition probabilities $q(\boldsymbol{x}_t \mid \boldsymbol{x}_{t-1})$, for $t = 1, \ldots, T$. Then a parametrized reverse process $p_\theta(\boldsymbol{x}_{t-1} \mid \boldsymbol{x}_t)$ is learned to invert the corruption $\boldsymbol{x}_T \to \boldsymbol{x}_0$ and recover samples from the original data distribution.

In masked diffusion models (Nie et al., 2025; Sahoo et al., 2024), the forward process consists of an absorbing noising process in which tokens are independently replaced by a special [MASK] token. Technically, the various sequences $\boldsymbol{x}_t$, for $t = 1, \ldots, T$, are latent variables and represented as categorical distributions over $\mathcal{V}$; when masked, the corresponding token is represented by a one-hot vector for the [MASK] token, denoted M in this paper.

Formally, the noising distribution governing the forward process from $\boldsymbol{x}_0 \to \boldsymbol{x}_T$ is defined as:

$$q(\boldsymbol{x}_t^i \mid \boldsymbol{x}_{t-1}^i) = \mathrm{Cat}\left(\boldsymbol{x}_t^i;\ \alpha_{t|t-1}\boldsymbol{x}_{t-1}^i + (1 - \alpha_{t|t-1})\mathrm{M}\right), \quad (1)$$

for each position $i \in [L]$ of the sequence where $\alpha_{t|t-1} := \frac{\alpha_t}{\alpha_{t-1}}$. Therein, $\{\alpha_t\}_{t=1}^T$ is a monotonically decreasing noise schedule with $\alpha_0 = 1$ and $\alpha_T = 0$ and $\mathrm{Cat}(\cdot; \boldsymbol{p})$ denotes the categorical distribution with probability vector $\boldsymbol{p}$. Essentially, for each position $i$, if the token is not already masked at time $t - 1$, it is preserved with probability $\alpha_{t|t-1}$ and replaced by [MASK] with probability $1 - \alpha_{t|t-1}$; once masked, it remains masked.

During generation, $\boldsymbol{x}_0$ is unknown, therefore, the reverse process is parameterized by a neural denoiser $\boldsymbol{x}_\theta(\boldsymbol{x}_t, t)$, typically implemented as a larger transformer network, that predicts a distribution over clean tokens, approximating $\boldsymbol{x}_0$. The reverse transitions from $\boldsymbol{x}_T \to \boldsymbol{x}_0$ are then given by:

$$p_\theta(x_{t-1}^i \mid \boldsymbol{x}_t) = \begin{cases} \mathrm{Cat}(x_{t-1}^i; x_t^i), & \text{if } x_t^i \neq \mathrm{M}, \\ \mathrm{Cat}\left(x_{t-1}^i; \gamma_t \mathrm{M} + \eta_t \boldsymbol{x}_\theta(x_t^i, t)\right), & \text{if } x_t^i = \mathrm{M} \end{cases} \quad (2)$$

for all $t = T, \ldots, 1$ and positions $i \in [L]$. Therein, the coefficients $\gamma_t = \frac{1 - \alpha_{t-1}}{1 - \alpha_t}$ and $\eta_t = \frac{\alpha_{t-1} - \alpha_t}{1 - \alpha_t}$ control the contribution of the denoiser prediction and the masking token in the reverse transition.

Once trained, sampling proceeds by initializing $\boldsymbol{x}_T$ as the fully masked sequence and iteratively denoises each step $T \to 0$ according to Equation (2). The final sample $\boldsymbol{x}_0$ is then returned as the generated sequence.

The task of discrete diffusion models is to maximize fidelity w.r.t. the original data distribution $p_{\mathrm{data}}(\boldsymbol{x}_0)$, but this does not, natively, enable a notion of control, which is central for scientific applications of interest to this paper. The next section formalizes the problem of constrained generation in discrete diffusion models.

## 4. Problem Formulation

The key desiderata of constrained generation is to produce sequences that lie in high-density regions of the data distribution while also satisfy a set of user-defined constraints $\mathcal{C}$. Formally, we seek to design a sampler $p_\theta^{\mathcal{C}}$ that meets the following objective:

Given some context $\boldsymbol{s}$, the sampler should produce samples $\boldsymbol{x}$ from the constrained target distribution:

$$p_\theta^{\mathcal{C}}(\boldsymbol{x}|\boldsymbol{s}) \propto p_\theta(\boldsymbol{x}|\boldsymbol{s}) \ \text{ subject to: } \ \boldsymbol{x} \in \mathcal{C}, \quad (3)$$

where $p_\theta(\boldsymbol{x}|\boldsymbol{s})$ is the base discrete diffusion model's learned distribution and $\mathcal{C}$ is the feasible region induced by a set of constraints.

These constraints can encode both discrete logical requirements (e.g., structural validity) and quantitative conditions (e.g., property scores exceeding a threshold), and the setting assumes access to a *constraint-violation function* that quantifies the extent to which a candidate sequence violates the constraints. Formally, let $\nu : \mathcal{V}^L \to \mathbb{R}_+^m$ be a vector-valued violation map, where $\nu_i(\boldsymbol{x})$ measures the violation of constraint $i$ and $\nu_i(\boldsymbol{x}) = 0$ indicates satisfaction. A sequence is feasible if and only if it incurs zero violation across all constraints, that is, $\boldsymbol{x} \in \mathcal{C}$ if $\nu(\boldsymbol{x}) = \boldsymbol{0}$.

Next the paper devises a training-free control mechanism for masked diffusion that induces a constraint-aware distribution $p_\theta^{\mathcal{C}}$ satisfying two desiderata: **(1)** *generative fidelity* and **(2)** *constraint feasibility*.

## 5. SearchDiff

**Method overview.** SearchDiff augments masked discrete diffusion with an inference-time search step that reduces constraint violation. Starting from the fully masked sequence $\boldsymbol{x}_T = (\mathrm{M}, \ldots, \mathrm{M})$, the reverse chain iterates for $t = T, \ldots, 1$; at each step, given the current latent state $\boldsymbol{x}_t \in \Delta^{|\mathcal{V}| \times L}$, where $\Delta$ denotes the probability simplex, the method performs the following three operations:

1. **Clean-state proposal.** The denoiser produces a per-position categorical distribution over the clean sequence,

$$\hat{\boldsymbol{x}}_0^{(t)} = \boldsymbol{x}_\theta(\boldsymbol{x}_t, t) \in \Delta^{|\mathcal{V}| \times L}, \quad (4)$$

which is interpreted as a proposal prior that is then used to generate clean sequences consistent with $\boldsymbol{x}_t$.

2. **Refine the proposal via constraint-aware search.** Using the denoiser output $\hat{\boldsymbol{x}}_0^{(t)}$ and the current diffusion context $(\boldsymbol{x}_t, t)$, SearchDiff invokes a discrete search operator that returns a refined discrete candidate $\bar{\boldsymbol{x}}_t$:

$$\bar{\boldsymbol{x}}_t \leftarrow \mathrm{SEARCH}(\hat{\boldsymbol{x}}_0^{(t)}, \boldsymbol{x}_t, t;\ \mathcal{C}). \quad (5)$$

The operator seeks to reduce constraint violation while remaining close to the original proposal. Importantly, the operator relies only on constraint evaluations in $\mathcal{C}$, without any gradient assumption.

3. **Sample and commit the next latent state.** Finally, it advances the diffusion chain using a modified reverse update: tokens that are already unmasked are set deterministically, while each currently masked position

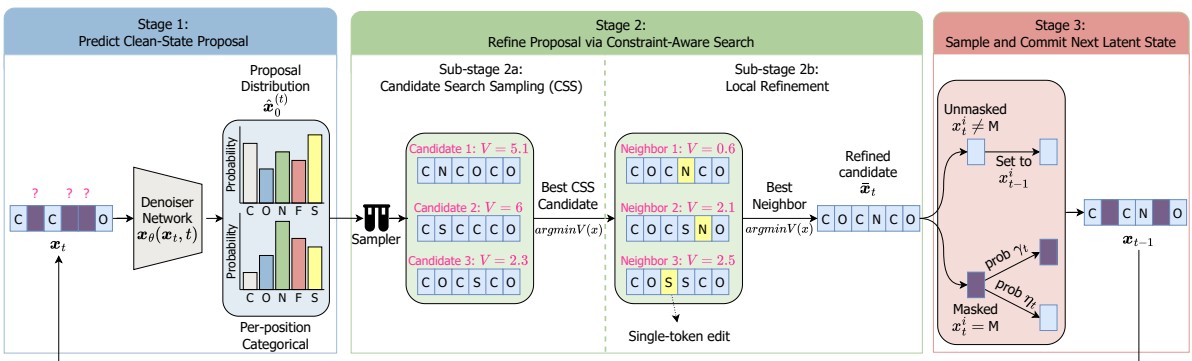

**Figure 2.** Illustration of a single SearchDiff denoising step: **(1)** The denoiser predicts a proposal distribution $\hat{x}_0^{(t)}$. **(2)** The search operator refines this into a low-violation candidate $\bar{x}_t$ via (2a) CSS and (2b) a local search refinement step. **(3)** The modified reverse kernel used $\bar{x}_t$ to deterministically update unmasked tokens and stochastically unmask a subset of masked tokens, yielding the next latent $x_{t-1}$.

samples a binary unmasking decision. If a position is chosen to unmask, its value is set to the token proposed by $\bar{x}_t$; otherwise it remains [MASK].

The full sampling trajectory can be summarized as:

$$x_t \xrightarrow{x_\theta(\cdot,t)} \hat{x}_0^{(t)} \xrightarrow{\text{SEARCH}(\cdot;\mathcal{C})} \bar{x}_t \xrightarrow{\bar{p}_\theta(x_{t-1}|\bar{x}_t,x_t)} x_{t-1},$$

for each $t = T, \ldots, 1$. Next, the paper formalizes the search operator of Step (2) and the sampling process of Step (3).

**5.1. Search-augmented denoising (Step 2)**

The denoiser $x_\theta$ defines a proposal distribution over clean sequences at each reverse step $t$, but it does not, on its own, enforce user-defined constraints. SearchDiff addresses this by interleaving discrete search moves with the denoiser prediction and the reverse transition. At step $t$, the search has two goals: **(1)** to produce a discrete candidate $\bar{x}_t$ with low constraint violation, and **(2)** to stay compatible with the denoiser's proposal so that the reverse transition follows the learned diffusion dynamics. An illustration of a single search-augmented denoising step is provided in Figure 2.

**Search problem.** At timestep $t$, SearchDiff defines a step-conditioned search problem

$$\mathcal{S}_t = \langle \mathcal{X}_t, \mathcal{A}_t, \text{Succ}_t, V \rangle,$$

where the diffusion context $(x_t, t)$ and denoiser proposal $\hat{x}_0^{(t)}$ are treated as fixed inputs. Therein, $\mathcal{X}_t$ is the state space of admissible candidates, $\mathcal{A}_t$ is the action space, $\text{Succ}_t$ is the successor function, and $V : \mathcal{V}^L \to \mathbb{R}_+$ is an aggregate constraint violation function (as defined below). The output of the search is a discrete candidate $\bar{x}_t \in \mathcal{V}^L$ that is used as input for the modified reverse kernel $\bar{p}_\theta$.

*Search space.* At reverse step $t$, the search operator optimizes over fully specified sequences $x \in \mathcal{V}^L$, notably not

incorporating the [MASK] token as unmasking is governed by the diffusion process rather than produced by search moves. The search is guided by the current diffusion context $(x_t, t)$ and the denoiser proposal $\hat{x}_{0,t}$, and returns a discrete candidate $\bar{x}_t$ that reduces constraint violation while remaining compatible with the model's proposal distribution.

*Actions and successor function.* Actions correspond to single-token edits. Given a candidate $x \in \mathcal{X}_t$, an action selects a position $i$ and a token $v \in \mathcal{V}$ and assigns $x^i \leftarrow v$:

$$\mathcal{A}_t(x) = \{(i,v) : v \in \mathcal{V}, \text{Succ}_t(x,(i,v)) \in \mathcal{X}_t\}. \quad (6)$$

where the successor function applies the edit and returns the resulting candidate,

$$\text{Succ}_t(x,(i,v)) = x' \quad \text{with} \quad x'^i = v, \ x'^j = x^j \ \forall j \neq i,$$

and rejects edits that violate admissibility.

*Objective function.* Consider a constraint set $\mathcal{C} = \{C_1, \ldots, C_K\}$; SearchDiff evaluates constraints through continuous violation functions $\nu_k$, each associated to a constraint $C_k$, as mentioned in Section 4. The constraint violation function is thus defined as the weighted sum of individual violations:

$$V(x) := \sum_{k=1}^K \lambda_k \nu_k(x), \quad (7)$$

where weights $\lambda_k \geq 0$ (defaulting to $\lambda_k = 1$) denote constraint priorities. The search operator computes a low-violation candidate at step $t$ by minimizing the objective:

$$\bar{x}_t \in \arg\min_{x \in \mathcal{X}_t} V(x). \quad (8)$$

The consistency requirement is enforced by the admissible set $\mathcal{X}_t$ and the denoiser output $\hat{x}_0^{(t)}$ is used as a proposal prior that biases the search toward candidates that remain plausible under the learned reverse dynamics.

## 5.2. SEARCH Initialization and local refinement

We instantiate SEARCH as a two-part procedure:

1. Obtain an initial candidate from the denoiser proposal;
2. Apply local refinements that decrease $V(\cdot)$.

**Candidate Search Sampling (CSS).** Note that, the input to the search operator includes the denoiser output $\hat{\boldsymbol{x}}_0^{(t)}$, which defines a distribution over clean sequences at step $t$. We leverage this distribution to construct a strong initial candidate via *Candidate Search Sampling* (CSS). Let $p_t(\cdot \mid \hat{\boldsymbol{x}}_0^{(t)}, \boldsymbol{x}_t)$ denote the proposal distribution over discrete candidates obtained by sampling tokens from $\hat{\boldsymbol{x}}_0^{(t)}$. Given a sampling budget $M$, CSS draws $\boldsymbol{x}_t^{(1)}, \dots, \boldsymbol{x}_t^{(M)} \sim p_t(\cdot \mid \hat{\boldsymbol{x}}_0^{(t)}, \boldsymbol{x}_t)$, and selects the least-violating sample:

$$\boldsymbol{x}_t^{\mathrm{css}} \in \underset{j \in \{1, \dots, M\}}{\arg\min} V(\boldsymbol{x}_t^{(j)}). \tag{9}$$

The selected $\boldsymbol{x}_t^{\mathrm{css}}$ is then used as the starting point for the search refinement; In Section 6 ablative results over CSS and refinement steps will illustrate their effects in isolation.

**Local search refinement.** Starting from $\boldsymbol{x}_t^{\mathrm{css}}$, the search process proceeds by performing iterative improvements. To ensure the problem complexity remains manageable, the process is instantiated using local search over single-token edits defined by the admissible action set (6). Let the step-conditioned Hamming-1 neighborhood be defined as

$$\mathcal{N}_t(\boldsymbol{x}) := \Big\{ \mathrm{Succ}_t(\boldsymbol{x}, (i, v)) : (i, v) \in \mathcal{A}_t(\boldsymbol{x}) \Big\}. \tag{10}$$

A single refinement round $r$ selects the best neighbor:

$$\boldsymbol{x}^{(r+1)} \in \underset{\boldsymbol{x}' \in \mathcal{N}_t(\boldsymbol{x}^{(r)})}{\arg\min} V(\boldsymbol{x}'). \tag{11}$$

The procedure iterates for $R$ rounds (or until no improvement is found), and returns the final refined candidate

$$\bar{\boldsymbol{x}}_t := \boldsymbol{x}^{(R)}. \tag{12}$$

## 5.3. Modified reverse kernel (Step 3)

Given the search-refined discrete proposal $\bar{\boldsymbol{x}}_t$, we advance the diffusion chain using a search-guided reverse update by sampling $\boldsymbol{x}_{t-1}$. This operates by stochastically unmasking positions. Concretely, the per-position reverse transition is

$$\bar{p}_\theta(x_{t-1}^i \mid \bar{\boldsymbol{x}}_t, \boldsymbol{x}_t) = \begin{cases} \mathrm{Cat}(x_{t-1}^i; \bar{x}_t^i), & \text{if } x_t^i \neq \mathtt{M}, \\ \mathrm{Cat}\left(x_{t-1}^i; \gamma_t \mathtt{M} + \eta_t \, \delta_{\bar{x}_t^i}\right), & \text{if } x_t^i = \mathtt{M}, \end{cases} \tag{13}$$

where $\delta_{\bar{x}_t^i}$ is a one-hot distribution that assigns probability 1 to the token chosen by search at position $i$.

A pseudocode summary of the full SearchDiff sampling procedure is provided in Algorithm 1 (Appendix).

# 6. Experiment

This section evaluates the performance of SearchDiff across a range of experimental settings, in the domains of biological sequence generation and symbolic reasoning SearchDiff is compared against state-of-the-art diffusion and autoregressive models of similar size.

## 6.1. Biological Sequence Generation

This section considers three inverse design biological sequence tasks: molecules, peptides, and transfer ribonucleic acid (tRNA). They share a common sequence-generation setup but differ in constraint structure and strictness. We compare SearchDiff against standard MDLM (Sahoo et al., 2024); guidance methods (CBG, CFG) for MDLM and UDLM as proposed by Schiff et al. (2024); autoregressive approaches (FUDGE (Yang & Klein, 2021), PPLM (Dathathri et al., 2019), and CFG (Sanchez et al., 2023)); CDD (Cardei et al., 2025); and an adapted TreeG-SC (Guo et al., 2025) implemented within the MDLM framework. Following prior work (Schiff et al., 2024), we report mean and variance over five runs, each generating 1024 sequences.

### 6.1.1. SMALL MOLECULES

**Settings.** Generating small molecules that are both chemically plausible and exhibit desirable properties is central to early-stage drug discovery, where invalid structures or hard-to-synthesize candidates can invalidate otherwise promising leads. We study constrained generation of SMILES strings under two property constraints: synthetic accessibility (SA), measuring ease of synthesis, and drug-likeness (QED), measuring agreement with common physicochemical profiles of successful drugs. In addition, we report standard quality metrics for *validity*, *novelty* (relative to the training set), and *uniqueness* within the generated batch. Both SA and QED are computed using non-differentiable black-box functions from RDKit[1] (see Appendix C.1.1). Following the setup in prior work (Schiff et al., 2024; Cardei et al., 2025), we use a 92.4M-parameter Transformer diffusion model trained on QM9 (Ramakrishnan et al., 2014); at inference, generation starts from an all-[MASK] sequence and runs 32 denoising steps to full unmasking, after which special tokens are replaced to obtain clean SMILES strings.

**Results.** Table 1 reports the number of admissible molecules, defined as valid, novel, and unique molecules, and the mean QED under QED maximization. SearchDiff produces $497.6 \pm 4.5$ admissible molecules, which is more than 3.5 times higher than the best-performing AR baseline (AR PPLM at $142.0 \pm 14.2$) and nearly 5 times higher than vanilla MDLM. It also achieves the highest QED, with mean 0.77 versus 0.62 for the next best method (UDLM CFG).

---

[1] https://www.rdkit.org/docs/api-docs.html

*Table 1.* Performance of different approaches on molecular generation under QED maximization task for molecular generation.

| Approach | Admissible Mol [↑] | Mean QED [↑] |
|---|---|---|
| MDLM | $397.8 \pm 12.9$ | $0.48 \pm 0.01$ |
| MDLM CBG | $116.6 \pm 8.9$ | $0.58 \pm 0.00$ |
| UDLM CBG | $63.8 \pm 8.1$ | $0.61 \pm 0.00$ |
| MDLM CFG | $95.8 \pm 9.0$ | $0.60 \pm 0.01$ |
| UDLM CFG | $64.0 \pm 5.1$ | $0.62 \pm 0.00$ |
| AR FUDGE | $53.0 \pm 3.5$ | $0.61 \pm 0.00$ |
| AR PPLM | $142.0 \pm 14.2$ | $0.45 \pm 0.00$ |
| AR CFG | $79.4 \pm 6.4$ | $0.60 \pm 0.00$ |
| SearchDiff | $\mathbf{497.6 \pm 4.5}$ | $\mathbf{0.77 \pm 0.00}$ |

*Table 2.* Performance of different approaches on molecular generation under (SA $\leq 4$, QED maximization) constraint for molecular generation. CDD results reported from (Cardei et al., 2025).

| Approach | Admissible Molecules [↑] | SA $\leq 4$ [↑] | Mean QED [↑] |
|---|---|---|---|
| MDLM | $397.8 \pm 12.9$ | $119.2 \pm 8.4$ | $0.48 \pm 0.01$ |
| CDD | 31 | 31 | 0.61 |
| TreeG-SC | $190.6 \pm 16.8$ | $150.2 \pm 17.0$ | $0.48 \pm 0.00$ |
| SearchDiff | $\mathbf{474.2 \pm 8.7}$ | $\mathbf{470.2 \pm 7.8}$ | $\mathbf{0.73 \pm 0.00}$ |

Notably, TreeG-SC struggles to balance validity and the SA constraint (Table 2). In contrast, SearchDiff preserves both: it produces $474.2$ admissible molecules, with $99\%$ of them meeting the strict SA $\leq 4$ threshold, while maintaining a strong mean QED of 0.73, higher than all baselines.

Next, we isolate the contributions of CSS and LS on the task of enforcing SA $\leq 4$ while maximizing QED (Table 3). Notice that increasing the CSS candidate budget improves results across settings, is the main driver of the gains and cannot be replicated by scaling CSS alone. In particular, SearchDiff with a single CSS candidate plus LS achieves $263.6$ molecules with SA $\leq 4$, substantially outperforming the variant with 128 CSS candidates but no LS (145.0). Applying search throughout the trajectory (all steps) further improves performance relative to post-hoc refinement applied only at the final step until convergence.

SearchDiff furthermore demonstrates strong out-of-distribution discovery results which is quantified by the number of novel molecules whose QED exceeds the maximum QED in the training set. As shown in Figure 3, LS is once again the key driver: note how standard decoding and the CBG produce no OOD candidates ($0.0 \pm 0.0$), and TreeG-SC remains negligible ($0.6 \pm 0.5$), whereas LS alone generates $212.6 \pm 11.7$ OOD molecules by applying local neighbor edits rather than following the model's global positional statistics. This refinement remains compatible with CSS: *the combined LS+CSS variant reaches an average of 332.4 molecules with higher QED than the training dataset*

*Table 3.* Ablation of SearchDiff search steps on QM9 dataset under constraints (SA $\leq 4$, maximize QED). Local search is either deactivated (top), active only in the last step, and run until convergence (middle), or active in each denoising step.

| Local search | # CSS cand. | SA $\leq 4$ [↑] | QED [↑] |
|---|---|---|---|
| **Not active** | 1 | 110.0 | 0.48 |
| | 32 | 143.2 | 0.62 |
| | 128 | 145.0 | 0.65 |
| **Last step until convergence** | 1 | 419.2 | 0.66 |
| | 32 | 456.2 | 0.72 |
| | 128 | 500.2 | 0.73 |
| **All steps** | 1 | 433.8 | 0.67 |
| | 32 | 470.2 | 0.73 |
| | 128 | **514.2** | **0.75** |

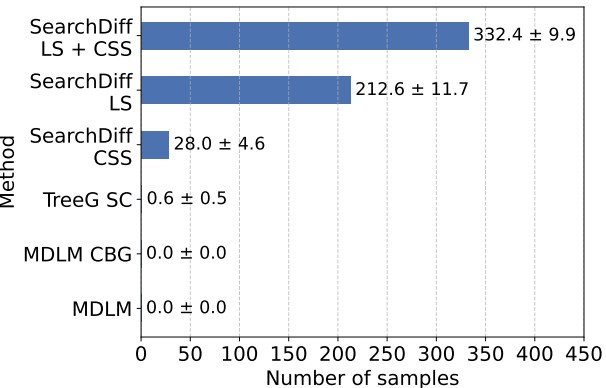

*Figure 3.* Average number of generated molecules with higher QED compared to the maximum value of QED in the training dataset for different algorithms.

*maximum value*, thus local refinement is crucial for exploring chemical regions not observed during training.

### 6.1.2. PEPTIDES

**Settings.** Peptide generation (Wan et al., 2022) aims to design short amino-acid sequences with properties that make them viable candidates for downstream synthesis and biological activity, where even small violations of biophysical requirements can render a sequence ineffective. This setting imposes three constraints for antimicrobial and host-defense peptides (Hancock & Sahl, 2006): sequence length between 10 and 50 amino acids, net positive charge in $[+2, +9]$ to promote interaction with negatively charged membranes, and hydrophobic residue fraction at least $30\%$ to support membrane insertion. These properties are calculated using non-differentiable black-box functions provided by `Biopython`[2] library (see Appendix C.1.2 for details). The base model uses the same architecture as in the molecular setting (Schiff et al., 2024); generation runs for 32

---

[2] https://biopython.org/docs/1.76/api/Bio.SeqUtils.ProtParam.html

*Table 4.* Performance of different approaches on peptide and tRNA generation in terms of numbers of unique and novel molecules satisfied these constraints. 1) **Peptide**: constraint 1: length $\leq 50$; constraint 2: hydrophobic ratio $\geq 30\%$; constraint 3: $2 \leq$ net charge $\leq 9$. 2) **tRNA**: constraint 1: $7 \leq$ acceptor stem length $\leq 9$; constraint 2: T$\Psi$C loop; constraint 3: D loop features a 4 to 6 base-pair stem, 5-base-pair stem enclosing the anticodon triplet, variable loop varies widely in length (3 to 21 bases).

| Approach | All constraints [↑] | Novel & unique [↑] | C1 Len $\leq 50$ [↑] | C2 Hydrophobic $\geq 30\%$ [↑] | C3 $2 \leq$ Charge $\leq 9$ [↑] |
|---|---|---|---|---|---|
| **Peptide** | | | | | |
| MDLM | $530.8 \pm 9.9$ | $679.2 \pm 8.2$ | $742.8 \pm 7.2$ | $549.0 \pm 16.3$ | $624.4 \pm 9.4$ |
| MDLM CBG | $471.6 \pm 19.6$ | $934.0 \pm 6.4$ | $712.0 \pm 18.7$ | $806.8 \pm 5.6$ | $572.2 \pm 16.6$ |
| UDLM CBG | $625.6 \pm 12.6$ | $795.4 \pm 17.4$ | $726.4 \pm 12.2$ | $743.8 \pm 14.8$ | $685.4 \pm 16.5$ |
| TreeG-SC | $867.2 \pm 4.5$ | $979.2 \pm 4.2$ | $979.2 \pm 4.2$ | $977.0 \pm 4.8$ | $867.6 \pm 4.2$ |
| SearchDiff | $\mathbf{1024 \pm 0.0}$ | $\mathbf{1024 \pm 0.0}$ | $\mathbf{1024 \pm 0.0}$ | $\mathbf{1024 \pm 0.0}$ | $\mathbf{1024 \pm 0.0}$ |

| Approach | All constraints [↑] | Novel & unique [↑] | C1 Acceptor stem [↑] | C2 T$\Psi$C loop [↑] | C3 D/anticodon/variable [↑] |
|---|---|---|---|---|---|
| **tRNA** | | | | | |
| MDLM | $25.4 \pm 5.5$ | $\mathbf{1005.2 \pm 4.1}$ | $445.4 \pm 16.0$ | $807.6 \pm 10.1$ | $46.2 \pm 4.7$ |
| MDLM CBG | $17.2 \pm 4.8$ | $942.0 \pm 4.7$ | $374.8 \pm 13.1$ | $744.8 \pm 12.5$ | $31.4 \pm 6.5$ |
| UDLM CBG | $20.8 \pm 4.3$ | $252.4 \pm 11.8$ | $199.4 \pm 14.4$ | $225.0 \pm 8.6$ | $34.6 \pm 3.2$ |
| TreeG-SC | $25.8 \pm 1.4$ | $730.4 \pm 11.2$ | $599.4 \pm 9.9$ | $694.2 \pm 11.5$ | $31.4 \pm 2.2$ |
| SearchDiff | $\mathbf{719.0 \pm 105.1}$ | $966.6 \pm 46.2$ | $\mathbf{964.6 \pm 45.8}$ | $\mathbf{955.8 \pm 51.4}$ | $\mathbf{726.0 \pm 102.3}$ |

**Results.** As shown in Table 4, SearchDiff attains a 0% constraint violation rate, generating 1024 peptides that are simultaneously valid, unique, novel, and satisfy *all peptide constraints*. This is in direct contrast to state of the art baselines such as TreeG-SC and UDLM CBG, which may perform well on individual criteria, but do not reliably satisfy the full constraint set. SearchDiff's strong performance across metrics follows from directly optimizing a black-box, gradient-free constraint evaluator through discrete edits during denoising, whereas gradient-based guidance relies on differentiable surrogates and vanilla MDLM sampling has no mechanism to repair violations.

### 6.1.3. TRANSFER RIBONUCLEIC ACID

**Settings.** Finally, we study structure-aware design for transfer RNA (tRNA). tRNAs are central adaptor molecules in translation function depends on conserved motifs and a characteristic stem–loop secondary structure. The task considers constraints that encode the core tRNA components: an acceptor stem with a conserved CCA tail (Itoh et al., 2013; Jahn et al., 1991), the D loop (Itoh et al., 2013), the anticodon stem–loop responsible for codon recognition (Itoh et al., 2013), the T$\Psi$C loop containing the conserved T$\Psi$C motif (Chan et al., 2013), and the variable loop whose length distinguishes different tRNA classes (Prabhakar et al., 2022; Brennan & Sundaralingam, 1976). These properties are calculated using non-differentiable black-box functions pro-

vided by ViennaRNA[3] (see Appendix C.1.3 for details). Importantly, these structural constraints ensure that generated sequences remain biologically plausible. We train the same base model as in the molecular setting (Schiff et al., 2024) and generate with 32 denoising steps and maximum length 110. However, compared to peptides and molecules, tRNA generation poses a substantially greater challenge due to the larger number of tokens that must be predicted at each denoising step under tightly coupled structural constraints.

**Results.** Table 4 tabulates the results for all models. Note that the base model MDLM performs rather poorly on this task while other baselines offer only limited gains. In contrast, SearchDiff yields a substantial improvement in joint feasibility achieving a $27\times$ increase in the "all constraints" metric relative to the best-performing baseline. The dominant failure mode across methods is constraint **C3**, which requires satisfying coupled structural features such as specific base-paired stems and variable-loop length, and thus depends on long-range dependencies that are both rare (OOD) and sharply defined. SearchDiff instead attains 726.0 on this constraint. Observing the produced traces we notice that local search consistently steers intermediate denoising states toward sequences whose edits repair structural violations and thus recover feasible folding configurations in the high-dimensional discrete space. An example of a valid tRNA generated by SearchDiff and an invalid tRNA generated by MDLM is shown in Figure 4.

---

[3] https://www.tbi.univie.ac.at/RNA

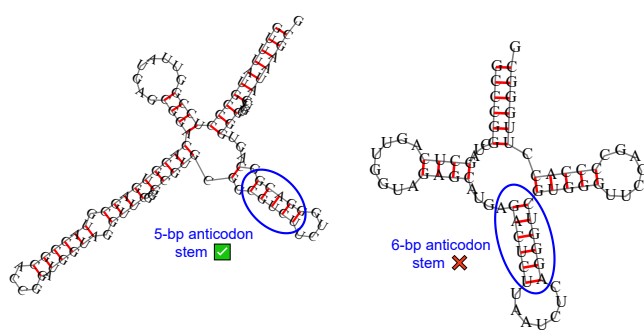

*Figure 4.* tRNAs generated by SearchDiff (left) and MDLM (right).

| Model | Size | Accuracy (%) [↑] | |
|---|---|---|---|
| | | **Sudoku** | **Boolean SAT** |
| AR (GPT-2) | 6M | 31.25 | 86.13 |
| AR (GPT-2) | 85M | 26.56 | 93.97 |
| AR (GPT-2) | 303M | 23.21 | **94.42** |
| MDM (Vanilla) | 6M | 93.2 | 9.6 |
| MDM (Last_Step) | 6M | 92.2 | 11.4 |
| MDM (SearchDiff) | 6M | **96.5** | 76.0 |

*Table 5.* Accuracy (%) across Sudoku (left) and Boolean Sat (right) 7 variables. For MDM, we report Vanilla sampling, SearchDiff, and the search only implemented on the last denoising step .

## 6.2. Symbolic Reasoning Tasks

**Settings.** Next, this section evaluates SearchDiff on two symbolic logic benchmarks, Sudoku and Boolean satisfiability, to assess constrained generation in regimes where feasibility is binary and constraints are global, non-differentiable, and tightly coupled. In both tasks, a single token error can simultaneously violate multiple invariants, so standard stochastic unmasking often fails to reach a feasible region. The settings use the same masked discrete diffusion model (MDM) backbone for both problems, with a GPT-2 style Transformer denoiser (causal mask removed for bidirectional conditioning) trained with the standard masked diffusion objective (Ye et al., 2024). For Sudoku, we represent a $9 \times 9$ board as a length-81 token sequence and format each instance as PUZZLE[SEP]SOLUTION, where PUZZLE fixes the givens and SOLUTION is generated by initializing target positions to [MASK] and denoising for $T$ steps. For Boolean SAT, we consider random 3-SAT with $n = 7$ variables, each with 45 clauses, and format each instance as FORMULA[SEP]ASSIGNMENT, where the model generates the length-$n$ assignment conditioned on the formula (Ye et al., 2024). In both cases, SearchDiff uses a black-box violation evaluator to guide discrete local edits: for Sudoku, it counts duplicates across rows, columns, and $3\times3$ subgrids; for SAT, it counts unsatisfied clauses. The neighborhood is defined by single-token replacements (cell value changes for Sudoku; truth-value flips for SAT), holding the conditioning prefix fixed. Additional experimental details are available in Appendix C.2.

**Results.** Table 5 illustrates accuracy for Sudoku (left) and Boolean SAT (right). For the Sudoku task, SearchDiff improves over vanilla MDM decoding (96.5% vs. 93.2%), while applying search only at the final step results in a weaker performance (92.2%) than the vanilla MDM, indicating that incorporating search throughout the denoising trajectory is preferable to post-hoc repair.

The Boolean SAT application is particularly revealing as the generation target is an assignment that must satisfy a set of coupled, global constraints, and there is essentially

no notion of "almost correct": flipping a single variable can simultaneously break many clauses, while the model receives no incremental signal about which bit caused the failure. In other words, the feasible set is a sparse, combinatorial subset of $\{0, 1\}^n$ and satisfaction is a discontinuous predicate, so stochastic unmasking can easily drift into large infeasible regions and has little structure to follow back toward feasibility. This makes the gap substantially larger: vanilla MDM achieves only 9.6% accuracy, while SearchDiff reaches 76.0% (a $\sim 7.9\times$ improvement), with last-step-only search again providing only marginal gains (11.4%). Notably, SearchDiff attains these results with a 6M-parameter diffusion model; larger GPT-2 autoregressive baselines do not close the gap on Sudoku and, more importantly, do not provide a comparable mechanism for enforcing hard constraints during sampling. Additional results across denoising step counts are reported in Appendix B.2.

## 7. Conclusion

This work introduced SearchDiff, a training-free, search-augmented framework for precise constrained generation with discrete diffusion models. The proposed framework interleaves candidate search sampling and iterative local refinement directly into the reverse denoising process of masked diffusion models. This results in a process that steers the pre-trained model distribution toward feasibility on the discrete constraint manifold, without incurring the overhead of additional training. The performance of SearchDiff was examined across a range of scientific and reasoning benchmarks, spanning biological sequence design and symbolic reasoning. Across all experiments, SearchDiff consistently improves constraint satisfaction and data faithfulness, while remaining computationally practical for discovery workflows. More broadly, these results suggest that trajectory-level, inference-time optimization may be a principled route to making discrete generative models more reliable and controllable under global, non-differentiable constraints in both scientific and reasoning tasks.

## Impact Statement

This paper presents work whose goal is to advance the field of Machine Learning. There are many potential societal consequences of our work, none which we feel must be specifically highlighted here.

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

## A. SearchDiff Pseudocode

Algorithm 1 (SearchDiff) augments the standard reverse chain of a masked discrete diffusion model (MDM) with an inference-time search step that reduces a user-specified constraint-violation score. The inputs are a pre-trained denoiser $\boldsymbol{x}_\theta$, a violation function $V(\cdot)$ that evaluates a fully discrete candidate sequence, the number of reverse steps $T$, a candidate search sampling (CSS) budget $M$, and a maximum number of local-refinement rounds $R$. The algorithm returns the final denoised sequence $\boldsymbol{x}_0$.

The procedure initializes the chain at the fully masked state $\boldsymbol{x}_T = (\texttt{M}, \ldots, \texttt{M})$ and iterates backward for $t = T, T-1, \ldots, 1$. At each step, the denoiser produces a clean prediction $\hat{\boldsymbol{x}}_0^{(t)} \leftarrow x\theta(\boldsymbol{x}_t, t)$, which serves as a guide for proposing improved intermediate states at level $t$. SearchDiff then performs search initialization via CSS: it draws $M$ candidates $\boldsymbol{x}_t^{(j)} \sim p_t(\cdot \mid \hat{\boldsymbol{x}}_0^{(t)}, \boldsymbol{x}_t)$ from the model's reverse-time proposal distribution at step $t$. Importantly, already-unmasked positions are clamped (positions with $x_t^i \neq \texttt{M}$ are preserved), so CSS only explores values for the currently masked coordinates. Among the $M$ candidates, the algorithm selects the lowest-violation sample

$$\boldsymbol{x}_t^{\text{css}} \leftarrow \arg \min_{j \in \{1, \ldots, M\}} V(\boldsymbol{x}_t^{(j)}),$$

which provides a "best-of-$M$" initialization for refinement.

Next, SearchDiff applies a greedy local search refinement starting from $\boldsymbol{x}_t^{\text{css}}$. For up to $R$ rounds, it searches the Hamming-1 neighborhood $\mathcal{N}_t(\boldsymbol{x}^{(r)})$ (all sequences that differ from $\boldsymbol{x}^{(r)}$ at exactly one position, with the neighborhood restricted to the editable coordinates at time $t$) and chooses the neighbor with minimal violation:

$$\boldsymbol{x}' \leftarrow \arg \min_{\tilde{\boldsymbol{x}} \in \mathcal{N}_t(\boldsymbol{x}^{(r)})} V(\tilde{\boldsymbol{x}}).$$

If this move strictly improves the violation, that is $V(\boldsymbol{x}') < V(\boldsymbol{x}^{(r)})$, the algorithm accepts it and continues; otherwise it terminates. The refined sequence after the last accepted move is denoted $\bar{\boldsymbol{x}}_t$.

Finally, SearchDiff uses $\bar{\boldsymbol{x}}_t$ to define a modified reverse kernel that preserves feasibility improvements while maintaining the diffusion-style stochastic unmasking. For each position $i \in 1, \ldots, L$, if the coordinate was already fixed at time $t$ (i.e., $x_t^i \neq \texttt{M}$), SearchDiff deterministically carries forward the refined value, $x_{t-1}^i \leftarrow \bar{x}_t^i$. If instead the coordinate is still masked, it samples from a two-component categorical distribution that either keeps the mask (with weight $\gamma_t$) or commits to the refined token $\bar{x}_t^i$ (with weight $\eta_t$), namely

$$x_{t-1}^i \sim \text{Cat}\left(\gamma_t \, \texttt{M} + \eta_t \, \delta_{\bar{x}_t^i}\right).$$

This step implements controlled unmasking guided by the refined candidate, ensuring that improvements found by CSS and local search influence the subsequent reverse trajectory rather than being applied only as a post-hoc repair. Repeating this process down to $t = 1$ yields the final discrete output $\boldsymbol{x}_0$.

---

**Algorithm 1** SearchDiff

---

**Input:** Pre-trained MDM denoiser $\boldsymbol{x}_\theta$, constraint-violation function $V(\cdot)$, steps $T$, CSS budget $M$, refinement rounds $R$.

**Output:** Clean discrete sequence $\boldsymbol{x}_0$

1: $\boldsymbol{x}_T \leftarrow (\texttt{M}, \ldots, \texttt{M})$ {Initialize fully masked}
2: **for** $t = T, T-1, \ldots, 1$ **do**
3:      $\hat{\boldsymbol{x}}_0^{(t)} \leftarrow \boldsymbol{x}_\theta(\boldsymbol{x}_t, t)$
4:      **Search Initialization via CSS:**
5:      **for** $j = 1, \ldots, M$ **do**
6:          Sample $\boldsymbol{x}_t^{(j)} \sim p_t(\cdot \mid \hat{\boldsymbol{x}}_0^{(t)}, \boldsymbol{x}_t)$ {Clamp positions with $x_t^i \neq \texttt{M}$}
7:      **end for**
8:      $\boldsymbol{x}_t^{\text{css}} \leftarrow \arg \min_{j \in \{1, \ldots, M\}} V(\boldsymbol{x}_t^{(j)})$
9:      **Local search refinement:**
10:     $\boldsymbol{x}^{(0)} \leftarrow \boldsymbol{x}_t^{\text{css}}$
11:     $r \leftarrow 0$
12:     **while** $r < R$ **do**
13:        $\boldsymbol{x}' \leftarrow \arg \min_{\tilde{\boldsymbol{x}} \in \mathcal{N}_t(\boldsymbol{x}^{(r)})} V(\tilde{\boldsymbol{x}})$ {$\mathcal{N}_t(\cdot)$ is the Hamming-1 neighborhood}
14:        **if** $V(\boldsymbol{x}') < V(\boldsymbol{x}^{(r)})$ **then**
15:          $\boldsymbol{x}^{(r+1)} \leftarrow \boldsymbol{x}'$ {Greedy improvement}
16:          $r \leftarrow r + 1$
17:        **else**
18:          **break** {Stop if no improvement}
19:        **end if**
20:     **end while**
21:     $\bar{\boldsymbol{x}}_t \leftarrow \boldsymbol{x}^{(r)}$
22:     **Modified reverse kernel:**
23:     **for** $i = 1, \ldots, L$ **do**
24:        **if** $x_t^i \neq \texttt{M}$ **then**
25:          $x_{t-1}^i \leftarrow \bar{x}_t^i$
26:        **else**
27:          Sample $x_{t-1}^i \sim \text{Cat}\left(\gamma_t \, \texttt{M} + \eta_t \, \delta_{\bar{x}_t^i}\right)$
28:        **end if**
29:     **end for**
30:     $\boldsymbol{x}_{t-1} \leftarrow (x_{t-1}^1, \ldots, x_{t-1}^L)$
31: **end for**
32: **return** $\boldsymbol{x}_0$

---

## B. Additional Results

### B.1. Biological Sequence Generation

As shown in Figure 5, SearchDiff exhibits a distinct test-time scaling phenomenon, where both variants show incremental performance gains as the number of CSS candidates increases. Specifically, the *No LS* variant scales from an

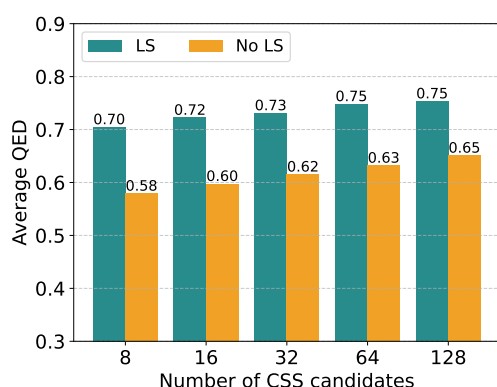

*Figure 5.* Average QED by LS and no LS approaches with different numbers of CSS candidates.

*Table 6.* Performance under increasing SA thresholds (comparison with CDD) on the QM9 dataset. CDD results reported from (Cardei et al., 2025).

| Method | SA satisfied | Mean QED |
|---|---|---|
| **SA $\leq 3.0$** | | |
| CDD | 36 | 0.63 |
| SearchDiff | $391.8 \pm 6.1$ | $0.66 \pm 0.01$ |
| **SA $\leq 3.5$** | | |
| CDD | 22 | 0.62 |
| SearchDiff | $448.6 \pm 9.0$ | $0.70 \pm 0.00$ |
| **SA $\leq 4.5$** | | |
| CDD | 33 | 0.58 |
| SearchDiff | $484.8 \pm 6.3$ | $0.75 \pm 0.00$ |

*Table 7.* Time per molecular generation of different SearchDiff's variations on QM9 dataset constraints (SA $\leq 4$, maximize QED).

| Method | Setting | Time (s) |
|---|---|---|
| MDLM | – | 0.67 |
| MDLM CBG | – | 1.28 |
| TreeG-SC | – | 32.12 |
| **SearchDiff** | | |
| Local search: No | 32 CSS candidates | 1.88 |
| | 128 CSS candidates | 6.00 |
| Local search: All steps | 1 CSS candidate | 3.66 |
| | 32 CSS candidates | 10.68 |

*Table 8.* Performance of different approaches on molecular generation in terms of number of valid, unique, and novel molecules satisfied these constraints: SA $\leq 3$ and QED $\geq 0.6$.

| Method | All constraints | SA $\leq 3$ | QED $\geq 0.6$ |
|---|---|---|---|
| MDLM | $1.2 \pm 0.7$ | $19.6 \pm 4.2$ | $4.8 \pm 1.9$ |
| TreeG-SC | $2.4 \pm 1.6$ | $66.6 \pm 5.5$ | $4.0 \pm 1.8$ |
| SearchDiff | $560.6 \pm 14.3$ | $682.8 \pm 9.0$ | $566.4 \pm 16.8$ |

average QED of 0.58 at 8 candidates to 0.65 at 128 candidates, while the *LS* variant improves from 0.70 to 0.75. This behavior indicates that the framework effectively leverages increased computational budget at inference time to explore a wider search space, consistently identifying higher-quality candidates across the discrete manifold.

To further investigate the flexibility of SearchDiff, we evaluate its performance against CDD across varying Synthetic Accessibility (SA) thresholds. As shown in Table 6, SearchDiff demonstrates a remarkable ability to adapt to both strict and relaxed constraints. While CDD remains limited to a very small set of satisfied samples (ranging from 22 to 36), SearchDiff consistently produces hundreds of satisfying molecules. SearchDiff also dominates CDD in terms of maximizing QED.

As shown in Table 7, all SearchDiff variations generate molecules faster compared to TreeG-SC while achieving superior results in configurations incorporating Local Search (LS), as evidenced in Table 3. Although utilising LS leads to increased computational time, it proves more efficient than high-volume sampling; notably, the variation with 128 CSS

candidates without LS is both slower and performs worse than variants using LS. While SearchDiff is still slower than vanilla MDLM, it remains a highly flexible, training-free method that can steer pre-trained models across diverse scientific use cases without the prohibitive costs of retraining or fine-tuning.

As shown in Table 8, we also consider the settings of constraints: SA $\leq 3$ and QED $\geq 0.6$ (Lee et al., 2025). The dramatic disparity in results shown in this table reveals that existing methods are predominantly confined to the high-density regions of the training data. For complex multi-objective tasks where $SA \leq 3$ and $QED \geq 0.6$, the feasible molecules are often located in rare or sparse regions of the chemical manifold, as shown in Figure 6 and Figure 7. MDLM and TreeG-SC primarily struggle to reach these solutions that reside in the "tails" of the training distribution. SearchDiff effectively overcomes this "distributional trap" by utilizing search-augmented steering to actively explore the discrete manifold. This capability enables it to successfully navigate into these rare feasible regions and identify hundreds of valid, unique, and novel molecules where traditional baselines find virtually none.

As shown in Figure 8, the number of molecules satisfying the stringent combined constraints of $SA \leq 3$ and $QED \geq 0.6$ underscores the test-time scaling capabilities of SearchDiff. Across all tested candidate counts, the *LS* variants consistently and significantly outperform the *No-LS* counterparts, achieving nearly triple the number of satisfied molecules even at the lowest search breadth. These results confirm that while increasing the number of CSS candidates

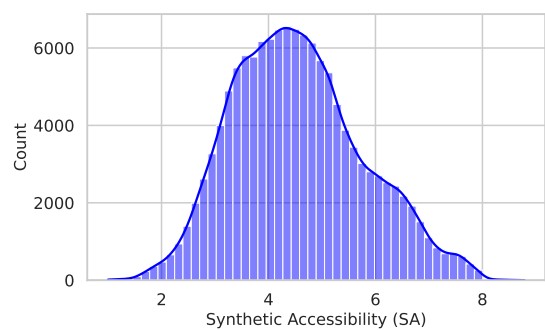

*Figure 6.* Distribution of Synthetic Accessibility (SA) scores in the training dataset.

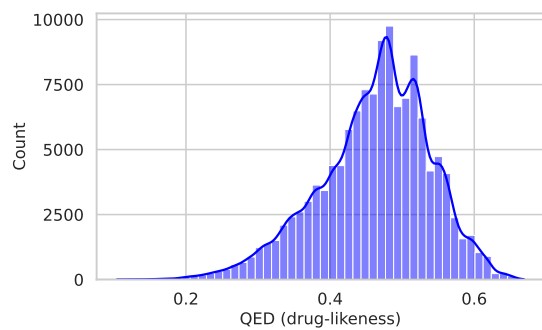

*Figure 7.* Distribution of QED (drug-likeness) scores in the training dataset.

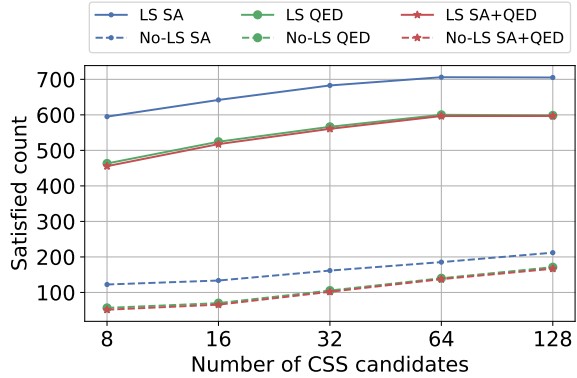

*Figure 8.* Number of satisfied molecules with different variants of SearchDiff and different numbers of CSS candidates versus other methods with constraints (SA $\leq$ 3, QED $\geq$ 0.6).

*Table 9.* Time per sequence generation of different methods for peptide and tRNA generation.

| Experiment | Method | Time (s) |
|---|---|---|
| **Peptides** | MDLM | 0.74 |
| | MDLM CBG | 1.66 |
| | TreeG-SC | 21.09 |
| | SearchDiff | 1.11 |
| **tRNA** | MDLM | 0.82 |
| | MDLM CBG | 2.49 |
| | TreeG-SC | 52.93 |
| | SearchDiff | 7.72 |

provides a steady improvement in results, the discrete local refinement provided by LS is the fundamental driver for successfully navigating complex, multi-dimensional constraint boundaries that are otherwise inaccessible through global sampling alone.

As shown in Table 9, SearchDiff achieves remarkable efficiency across both Peptides and tRNA generation tasks. In the Peptides experiment, SearchDiff (1.11s) introduces only a marginal overhead compared to the unconstrained MDLM (0.74s) and, notably, performs faster than the guided MDLM CBG baseline (1.66s). Compared to the grammar-based TreeG-SC (21.09s), SearchDiff is approximately 19× faster while providing superior constraint satisfaction.

For the more complex tRNA domain, although the latency increases to 7.72s (reflecting the computational cost of the unified folding simulation required for structural verification) SearchDiff still maintains a nearly 7× speed advantage over TreeG-SC (52.93s). These results demonstrate that our training-free framework is highly practical for large-scale scientific discovery where both structural integrity and inference speed are paramount.

## B.2. Symbolic Reasoning

Across both symbolic reasoning tasks, as observed in Table 10, varying the number of denoising steps $T$ highlights that SearchDiff's improvements are consistent across different amount of denoising steps in the reverse diffusion chains. For Sudoku, vanilla MDM improves steadily as $T$ increases (69.3→93.2), while SearchDiff remains consistently higher at every $T$ (91.4→96.5), with the largest absolute gain occurring with smaller $T$ value. This indicates that search-augmented refinement is particularly effective when the diffusion process has limited opportunity to correct early mistakes. For Boolean SAT, vanilla MDM stays consistently low across all $T$ (9.4–9.6), suggesting that additional denoising steps alone do not help the sampler locate the sparse feasible set; in contrast, SearchDiff achieves strong accuracy even at $T=1$ (71.5) and improves modestly with more steps (up to 76.0 at $T=20$), indicating that multi-step denoising mainly provides incremental benefit once search supplies a feasibility-driven direction. Finally, applying search only at the last step results in weaker performance across $T$ on both tasks, reinforcing that distributing search throughout the denoising trajectory is more effective than post-hoc repair.

*Table 10.* Accuracy (%) across tasks. For MDM, we report Vanilla sampling, SearchDiff, and SearchDiff applied to the last step under denoising steps $T \in \{1, 5, 10, 15, 20\}$.

| Model | Time Steps | Sudoku (Acc.%) | | | Boolean SAT (Acc.%, 7 vars) | | |
|---|---|---|---|---|---|---|---|
| | | Vanilla | SearchDiff | Last_Step | Vanilla | SearchDiff | Last_Step |
| | $T{=}1$ | 69.3 | 91.4 | - | 9.4 | 71.5 | - |
| | $T{=}5$ | 89.3 | 95.6 | 89.0 | 9.6 | 72.6 | 17.6 |
| **MDM (6M)** | $T{=}10$ | 91.9 | 95.9 | 92.1 | 9.5 | 72.6 | 11.7 |
| | $T{=}15$ | 93.4 | 96.7 | 92.4 | 9.5 | 73.8 | 11.5 |
| | $T{=}20$ | 93.2 | 96.5 | 92.2 | 9.6 | 76.0 | 11.4 |
| AR (GPT-2) | 6M | 31.25 | – | – | 86.13 | – | – |
| AR (GPT-2) | 85M | 26.56 | – | – | 93.97 | – | – |
| AR (GPT-2) | 303M | 23.21 | – | – | 94.42 | – | – |

## C. Experimental Settings

### C.1. Biological Sequence Generation

**Hardware Configuration** All experiments for biological sequence generation were conducted on NVIDIA A16 GPUs.

#### C.1.1. MOLECULAR GENERATION

Molecular generation refers to the use of computational models to create novel molecular structures with desired chemical and biological properties. This field plays a crucial role in drug discovery and material science by accelerating the design of compounds that meet specific criteria, such as synthetic asccessibility (SA) and drug-likeness (QED) (Cardei et al., 2025; Schiff et al., 2024).

We use the same settings as existing papers (Cardei et al., 2025; Schiff et al., 2024). We generate SMILES sequences using a base Transformer model of 92.4 million parameters, which was trained on the QM9 (Ramakrishnan et al., 2014) dataset. At inference time, the diffusion sampler initializes the target SMILES sequence to [MASK] and iteratively denoises over 32 steps until all target positions are unmasked. Given the fully unmasked sequences, all special tokens are replaced to achieve the clean SMILES representations. At each diffusion step, the predicted clean sequence is post-processed to remove unmatched branch parentheses and invalid ring indices before evaluation. This study uses the QM9 dataset (Ramakrishnan et al., 2014), which consists of small organic molecules with up to 9 heavy atoms.

This study focuses on generating molecules that satisfy key constraints, including chemical validity, novelty, SA, and QED. We use SA and QED as the evaluators of SearchDiff. At each diffusion step, we predict a clean SMILES sequence and apply a post-processing step that removes unmatched branch parentheses and invalid ring indices before feeding the sequence to the evaluators. For neighboring search, we consider all valid atom types supported in QM9. Ensuring these constraints is essential for producing molecules that

are not only chemically valid but also novel, practically synthesizable, and pharmaceutically relevant.

**Search Operator implementation.** To steer the denoising trajectory toward the feasible manifold, our search operator evaluates and ranks candidates based on their proximity to satisfying domain-specific constraints. Specifically, for each candidate molecule $x$, we compute its Synthetic Accessibility score and Quantitative Estimate of Drug-likeness. We quantify the distance of these properties to the satisfaction region; for instance, the SA violation is defined as $\max(0, SA(x) - \tau)$, where $\tau$ is the threshold for synthesizability. In our implementation, we adopt a hierarchical selection strategy that prioritizes SA satisfaction over QED optimization. When comparing two candidate molecules, the operator first selects the one with the lower SA violation; QED values are only considered as a secondary tie-breaker once a baseline level of synthetic accessibility is achieved. This prioritization ensures that SearchDiff fundamentally secures structural validity and chemical feasibility before attempting to refine more complex medicinal properties. To compute SA and QED scores, this study utilizes the widely adopted RDKit[4] library, which offers reliable implementations of these metrics. Ensuring these constraints is essential for producing molecules that are not only chemically valid but also novel, practically synthesizable, and pharmaceutically relevant.

#### C.1.2. PEPTIDES GENERATION

Peptide generation (Wan et al., 2022) involves designing and synthesizing short chains of amino acids with specific biological or chemical properties. Computational peptide generation aims to create novel peptide sequences that satisfy functional constraints such as antimicrobial activity, stability, or binding affinity. These peptides have broad applications in therapeutics, diagnostics, and biotechnology. Generative models help explore the vast combinatorial space

---

[4]https://www.rdkit.org/docs/api-docs.html

of possible sequences to identify candidates with desirable characteristics efficiently.

The constraints chosen for peptide generation: short sequences of 10 to 50 amino acids, an overall positive charge typically between +2 and +9, and a substantial proportion ($\geq$30%) of hydrophobic residues; are grounded in well-established biological principles of antimicrobial and host-defense peptides (Hancock & Sahl, 2006). These short cationic amphiphilic peptides play crucial roles in innate immune defense across nearly all life forms, acting as natural antibiotics with broad-spectrum antimicrobial activities. Their positive charge facilitates interaction with negatively charged microbial membranes, while the hydrophobic residues enable the formation of amphipathic structures critical for membrane disruption and antimicrobial function. By imposing these constraints, the generated peptides are more likely to adopt structural and functional properties characteristic of effective antimicrobial agents, enhancing their potential therapeutic relevance. Additionally, these properties have been shown to influence peptide stability, specificity, and resistance to degradation, all important factors for developing novel peptide-based drugs.

This study employs the GRAMPA dataset (Witten & Witten, 2019) to evaluate generation capabilities on peptide sequences. This is a robust database containing sequences and experimental Minimum Inhibitory Concentration (MIC) values of antimicrobial peptides (AMPs).

**Search Operator implementation.** The search operator steers the denoising trajectory by evaluating candidates based on their biological viability and functional stability. We quantify the distance to the feasible set using structural constraints, such as net charge or hydrophobic ratio, alongside the normalized folding energy ($\Delta G$) as the primary property for optimization. For a candidate sequence $x$, we calculate a violation score for each predefined constraint; for example, if a specific net charge is required, the violation is defined as $|charge(x) - target|$. The calculations were performed using `Biopython`[5] library.

### C.1.3. TRANSFER RIBONUCLEIC ACID GENERATION

Transfer ribonucleic acid (Rich & RajBhandary, 1976) (tRNA) generation involves creating synthetic or predicted tRNA sequences that mimic natural tRNAs' essential roles in protein synthesis. In designing constraints for tRNA sequence generation, it is essential to capture key structural elements critical for tRNA function. The acceptor stem typically consists of 7 to 9 base pairs formed by the pairing of the 5' and 3' terminal nucleotides, with the 3' end containing the conserved CCA tail for amino acid attach-

ment (Itoh et al., 2013; Jahn et al., 1991). This stem may include non-canonical base pairs, reflecting natural structural flexibility. The D loop features a 4 to 6 base-pair stem and commonly contains dihydrouridine modifications (Itoh et al., 2013). The anticodon loop, crucial for codon recognition, generally consists of a 5-base-pair stem enclosing the anticodon triplet (Itoh et al., 2013). The TΨC loop derives its name from the characteristic inclusion of pseudouridine (Ψ), a chemically modified form of uridine, within its sequence. This nucleotide frequently appears as part of a conserved motif written as 5'-TΨCGA-3', in which the ribothymidine forms a stabilizing base pair with the adjacent adenine (Chan et al., 2013). Additionally, the variable loop, located between the anticodon and TΨC loops, varies widely in length (3 to 21 bases) and can form a rigid arm in some tRNAs, differentiating class I and class II tRNAs (Prabhakar et al., 2022; Brennan & Sundaralingam, 1976). Incorporating these structural features as constraints ensures generated tRNA sequences maintain biological plausibility and functionality.

We trained the base model with the same network architecture as the molecular settings. The number of denoising steps remains 32, but the maximum sequence length is now 110. This makes the problem pose a significantly higher challenge than peptides and molecules due to the high average number of tokens to be unmasked per denoising step. For RNA modeling, this study curates a dataset of mammal tRNA sequences obtained from the GtRNAdb (Chan & Lowe, 2016) database.

**Search Operator implementation.** The search operator is designed to navigate the discrete manifold by prioritizing the preservation of the canonical secondary structure. We characterize the feasible set through a combination of structural constraints. For each candidate sequence, the operator calculates a violation score based on its sequence-structure consistency; for instance, a violation occurs if a predicted sequence fails to form the requisite hydrogen bonds necessary for stable folding. Specifically, we group the D-loop (4-6 base-pair stem), the 5-base-pair anticodon stem, and the variable loop (3-21 bases) into a single collective fold constraint. The rationale for this grouping is that all these constraints are derived from the simulated secondary structure, which must be computed before these specific topological values can be evaluated. By unifying them into a single operator, we achieve significant computational efficiency, as the secondary structure is simulated only once per candidate instead of three separate, redundant, and time-consuming simulations. The properties of tRNA sequences are computed using the `ViennaRNA`[6] package.

---

[5]https://biopython.org/docs/1.76/api/Bio.SeqUtils.ProtParam.html

[6]https://www.tbi.univie.ac.at/RNA

## C.2. Symbolic Reasoning

### C.2.1. SUDOKU

**Sudoku Search Operator.** Search is guided by a simple constraint-only evaluator that measures how far a candidate grid is from satisfying Sudoku rules. Given a predicted $9 \times 9$ board, the evaluator computes a violation score by counting duplicate digits in each of the 27 Sudoku units (9 rows, 9 columns, and 9 $3 \times 3$ subgrids). The total violation score is the sum across all units, where lower is better and a score of 0 indicates no duplicates anywhere. The search space consists of all candidate fillings of the Sudoku board on the SOLUTION side of the sequence. We consider local, token-level edits to the candidate grid: an action is a single-cell replacement that changes the digit at an editable position to an alternative value in $\{1, \ldots, 9\}$.

### C.2.2. BOOLEAN SATISFIABILITY PROBLEM

**SAT Search Operator.** Search is guided by a clause-based evaluator computed from the predicted assignment. Given a decoded candidate, it will parse the CNF clauses and evaluate each clause under the assignment. A clause satisfied if at least one of its literals evaluates to True; it is violated if all literals evaluate to False. The primary penalty is the number of violated clauses where lower is better. To prevent degenerate candidates it additionally penalizes undecided clauses and heavily penalize unassigned variables when present.

