# OpenReview forum: "Search-Augmented Masked Diffusion Models for Constrained Generation"
_ICML.cc/2026/Conference — Submitted to ICML 2026_

### Official Review · Reviewer_HGxR · 2026-03-07

**Soundness:** 3
**Presentation:** 3
**Significance:** 3
**Originality:** 3
**Overall Recommendation:** 5
**Confidence:** 4

**Summary:**

This paper proposes applying search according to user-defined objectives and/or constraints (which need not be differentiable) as part of each denoising step in masked discrete diffusion. This search can make edits to the denoiser’s output that improve the objective value or constraint satisfaction at each step of the reverse diffusion process. The effectiveness of the proposed algorithm is demonstrated on guided generation problems in a variety of settings: small molecules, peptides, tRNA as well as Sudoku solving and Boolean SAT.

**Compliance With Llm Reviewing Policy:**

Affirmed.

**Final Justification:**

I maintain my original recommendation to accept this paper. Authors provided additional clarifying experiments during the rebuttal period which has increased my confidence in the paper.

**Key Questions For Authors:**

* **Q1.** Might it be fairer to compare your method against the top-k sample produced by vanilla MDM, where k is set to match the number of CSS candidates? (Note that this is different from the last-step experiment you already include, as it would involve sampling k candidates independently rather than allowing corrections only at the last diffusion timestep.) Would it be possible to share results of such an experiment?
* **Q2.** For Sudoku, how can MDM with last step correction be worse than vanilla MDM (Table 5)?
* **Q3.** In practice, how much of a limitation is it that only Hamming-1 neighborhoods are considered during search? E.g. for Boolean SAT you mention that there is no notion of “almost correct”, so why do you think that your method can still succeed?
  * a. For example: in the case of Boolean SAT, your CSS procedure would seem to only work if it gets lucky and samples a valid candidate. If it gets lucky, then there is no need to run the refinement search. If it does not get lucky, then the CSS output will be an arbitrary sample, and the search-based refinement can only succeed if there is a Hamming-1 edit to fix the candidate. Can you comment on whether CSS or search refinement is more important in problems such as boolean SAT with binary-valued constraint violation functions?
  * b. Have you tried visualizing the number of corrections vs. diffusion timestep? I am curious whether certain points during the denoising process are more amenable to corrections, and how problem-dependent this structure is.
* **Q4.** How many rounds $R$ do you use in practice? How critical is this choice to performance?

**Limitations:**

No limitations are discussed in the current version of the paper. It may be worthwhile to discuss the reliance of the method on the small number of discrete values: for domains with a much larger number of possible token values (e.g. text), could the size of even the Hamming-1 neighborhood be large enough to make search impractical?

**Strengths And Weaknesses:**

The search-based method proposed in this paper is shown to be general and effective. This is motivated and demonstrated using several experiments that employ non-differentiable black box solvers as part of objective and constraint evaluation.

The flexibility afforded in specifying objectives and constraints makes the proposed method suitable for a wide range of applications. One potential limitation, however, is that the search-based correction mechanism appears to rely on the small size of the discrete vocabulary. With larger vocabularies in other domains (e.g., text), the size of the Hamming-1 neighborhoods that need to be considered at each search step might be quite large, requiring a potentially very expensive search step.

Overall, the paper is well structured and clearly written. Some minor notes on presentation:
* I did not see a substantial difference between figures 1 and 2. In my opinion, figure 2 is a clearer version of figure 1 and could fully replace it.
* There may be a few typos, e.g. in L161: “while also satisfy” –> while also satisfying

---

Note: I am quite unfamiliar with the biological domain. Therefore, while I found the biological sequence generation experiments convincing, I may not be able to fully judge their significance and soundness.

---

> ### Author Rebuttal · Authors · 2026-03-30
>
> Thank you for the helpful review, and especially for recognizing the generality and effectiveness of the method, its flexibility for black-box objectives/constraints, and the clarity of the presentation. We also appreciate the concrete suggestions. We will simplify the presentation by consolidating Figs. 1 and 2, fix the noted typo, and expand the limitations discussion on scaling to larger vocabularies.
>
> > **Q1. I did not see a substantial difference between figures 1 and 2. In my opinion, figure 2 is a clearer version of figure 1 and could fully replace it.**
>
> We agree this can be improved. Figure 1 was intended as a high-level intuition figure and Figure 2 was intended as the step-by-step technical view. In the revision, we will simplify this part of the presentation by consolidating the two. This will also allow us to use the gained space for discussing the results collected during rebuttal (see below).
>
> > **Q2. Might it be fairer to compare your method against the top-k sample produced by vanilla MDM, where k is set to match the number of CSS candidates?**
>
> Yes, this is a very good suggestion. We ran this comparison with $k=32$, matching the CSS budget:
>
> |Setting|Top-32 vanilla MDM|SearchDiff|
> |-|-|-|
> |SA ≤ 3, QED ≥ 0.6|40.6 ± 6.6|560.6 ± 14.3|
> |Mean QED (QED maximization task)|0.58 ± 0.00|0.77 ± 0.00|
> |# samples above train max QED|4.4 ± 0.8|332.4 ± 9.9|
>
> This isolates the effect of broader proposal sampling from the effect of local search and shows that the gains do not come simply from taking the best of more vanilla MDM samples.
>
> > **Q3. For Sudoku, how can MDM with last step correction be worse than vanilla MDM (Table 5)?**
>
> This is a fair question. The originally reported gap is very small; after rerunning with 5 seeds, it becomes even smaller:
>
> |Setting|Accuracy (%)|
> |-|-|
> |Vanilla MDM, $T=20$|$93.32 ± 0.91$|
> |MDM + Last-Step, $T=20$|$93.10 ± 0.64$|
> |SearchDiff, $T=20$|$97.38 ± 0.56$|
> |SearchDiff, $T=15$|$96.88 ± 0.64$|
> |SearchDiff, $T=10$|$96.36 ± 0.61$|
> |SearchDiff, $T=5$|$95.50 ± 0.47$|
> |SearchDiff, $T=1$|$91.84 ± 0.45$|
>
> So the vanilla-vs-last-step difference is only $0.22$ points and is within seed variability. Intuitively, a single post-hoc correction at the end has limited ability to repair globally coupled conflicts, whereas trajectory-level search can steer intermediate states earlier toward feasible boards. We will expand the description in the paper to report these results.
>
> > **Q4(a). How limiting is Hamming-1 search? For SAT, why can it still succeed?**
>
> We agree Hamming-1 search is less expressive than larger neighborhoods, but this is a deliberate tradeoff: it keeps each search step cheap enough to apply throughout denoising. Importantly, SearchDiff does not search from arbitrary random states, but from denoiser-guided candidates already biased toward plausible regions. Thus, Hamming-1 acts as a local correction mechanism. For SAT, while the final success metric is binary, the search signal is not: we minimize the number of unsatisfied clauses. This graded violation signal makes small local edits meaningful and allows improvements to accumulate across denoising steps. As noted in our response to Q1 with reviewer udfW, we only choose the best sample at each step to prevent the exponential cost. During rebuttal, we also tested multiple LS per step in the same molecular setting as Table 2, as noted in the response to ALkB in Q3.
>
>
> > **Q4(b). Have you tried visualizing the number of corrections vs. diffusion timestep?**
>
> Yes. On the full Sudoku test set (1000 instances, 5 seeds), corrections are concentrated early in the trajectory:
>
> |Step|Improvement rate (%)|Step|Improvement rate (%)|
> |-|-|-|-|
> |20|$9.04 ± 0.46$|10|$1.02 ± 0.16$|
> |19|$5.76 ± 0.69$|5|$0.30 ± 0.10$|
> |18|$4.52 ± 0.67$|1|$0.06 ± 0.05$|
> |17|$3.18 ± 0.99$| | |
> |15|$2.02 ± 0.53$| | |
>
> The full profile decreases nearly monotonically from early to late timesteps; thus early partially formed states are much more amenable to corrective edits, while later states are largely stabilized. We will add this diagnostic in the final paper and note that the exact profile is problem-dependent.
>
> > **Q5. How many rounds do you use in practice? How critical is this choice?**
>
> By default, each denoising step uses **one** local refinement. The global-local pattern is then repeated across timesteps. During rebuttal, we also tested multiple LS refinements per step in the same molecular setting as Table 2, as noted in the response to ALkB in Q3.
>
> Note that the gains beyond one refinement are modest, so the default choice captures most of the benefit while keeping cost low. We will discuss this design choice and its benefit explicitly in the revision.
>
> We thank the reviewer again for their thoughtful review. We believe these added experiments and clarifications strengthen the paper further and address the remaining concerns.

---

> > ### Author Rebuttal · Reviewer_HGxR · 2026-04-01
> >
> > I would like to thank the authors for their helpful responses and for running additional clarifying experiments. I will keep my recommendation to accept this paper.

---

> > > ### Author Response · Authors · 2026-04-03
> > >
> > > Thank you again for your careful reading and thoughtful engagement throughout the discussion. We sincerely appreciate your feedback and are glad that the additional experiments and clarifications helped address your concerns. We are grateful for your support of the paper!

---

### Official Review · Reviewer_5gFu · 2026-03-12

**Soundness:** 2
**Presentation:** 3
**Significance:** 3
**Originality:** 3
**Overall Recommendation:** 3
**Confidence:** 4

**Summary:**

This work introduces a training-free method for constrained generation with masked diffusion models. The goal is to sample from the constrained distribution, which is defined as a subset of the original data distribution that satisfies a set of constraints. The proposed method, SearchDiff, uses a two-stage search strategy in each denoising step to iteratively guide the generation process toward feasible regions. The first step is a pretrained model proposing a set of candidate samples in the clean-data space, and then the sample with the least constraint violation is selected. The second step is local search within a 1-Hamming ball of the selected sample to further refine the sample. Then, the following denoising step selects the index $i$ to unmask, replacing the mask token at $i$ with the $i$-th token of the searched sample. The method is evaluated on tasks requiring strict structural constraints, including biological sequence generation and symbolic reasoning.

**Compliance With Llm Reviewing Policy:**

Affirmed.

**Final Justification:**

There is a gap between what the authors aim to achieve with their "ideal target distribution" (eq. 3) and what they actually achieve empirically. The former is a Bayesian posterior of a pretrained generative model with the constraints as likelihood in the sample space, while the latter is a heuristic search with the model's structural prior at each denoising step. In their experiments, while the proposed method has empirically shown better performance in terms of constraint satisfaction, there lacks clear evidence that it actually achieved what the authors intended in the first place.

I recognise its empirical effectiveness, and I like the idea of exploiting a step-wise structural prior from a generative model. The reason I'm still hesitant to recommend acceptance is that the current manuscript is not well articulated in the sense I mentioned above, and I believe eq. 3 and related "ideal target distribution" claims should be revised to prevent confusion.

**Key Questions For Authors:**

1. I wonder if the proposed method guarantees sampling within the constrained set (I assume not, because of the results in Table 4). If not, could you provide any intuition on when and why the method fails?
2. I wonder if the evaluation is done in a fair way. Especially, did baselines also utilise the function $\nu$? Did the baselines used similar amount of computation? (This should depend on the bottleneck of each algorithm, so I hope the authors can give some details about it.)

*Note: Given that the questions are rather minor, I don't think my evaluation of this paper is affected by the answers to these questions. But addressing the major points in Weaknesses can definitely change my evaluation.*

**Limitations:**

While the paper includes an impact statement, it lacks a discussion of limitations.

**Strengths And Weaknesses:**

### Strengths

1. The writing is easy to follow.
2. The proposed idea is interesting and straightforward. It also showed promising empirical results against multiple baselines.

### Weaknesses

1. Most importantly, the method is not well-principled. While the goal is to sample from the constrained distribution (c.f., eq. (3)), samples from the proposed method do not guarantee achieving this. The proposed method is more of a heuristic for obtaining approximate samples from the distribution. The problem formulation should be fixed, and this point should be discussed in the main text.
2. Related to the first point, more principled methods that actually can sample from the constrained distribution exist [1, 2, 3, 4] if you properly define the reward-tilted target distribution, for example, $p_{r}(x) = p(x)r(x)$ where $r(x)=1$ if $x$ satisfies the constraint and $r(x)=0$ otherwise. One of the basic algorithms you can imagine is just doing rejection sampling with this as the target distribution and the pretrained model as the proposal distribution.
3. (minor) In line 151 left, the authors state that the discrete diffusion models maximise fidelity of the original data distribution, but actually, what those models maximise is the likelihood (or ELBO) of the original data distribution.

[1] Li, Xiner, et al. "Derivative-free guidance in continuous and discrete diffusion models with soft value-based decoding." NeurIPS 2025.
[2] Uehara, Masatoshi, et al. "Reward-Guided Iterative Refinement in Diffusion Models at Test-Time with Applications to Protein and DNA Design." ICML 2025.
[3] Chu, Wenda, et al. "Split gibbs discrete diffusion posterior sampling." NeurIPS 2025.
[4] Wu, Luhuan, et al. "Practical and asymptotically exact conditional sampling in diffusion models." NeurIPS 2023. (which is for continuous-space diffusion but can naturally be applied to discrete-space diffusion)

---

> ### Author Rebuttal · Authors · 2026-03-30
>
> We thank the reviewer for the careful reading and for noting that the paper is easy to follow, the core idea is interesting, and the empirical results are promising. We also agree that the framing around Eq. (3) can be improved and will make sure this aspect is captured in the revised paper.
>
> **On Eq. (3) and principled sampling.**
>
> We agree; Eq. (3) should be interpreted as an *ideal target distribution*, while SearchDiff is a *training-free approximate algorithm* that steers sampling toward feasible, high-quality regions using denoiser-guided search. It does *not* guarantee exact sampling from the constrained target. We will revise the paragraph following Eq. (3) to make this explicit.
> This distinction is especially important in our study since we also aim to reach "out-of-distribution" but feasible regions that have extremely low probability under the base model. In such settings, simple rejection sampling from the pretrained model is not a practical alternative. This issue is concrete in our OOD molecule experiments, where vanilla MDM finds such samples very rarely, whereas SearchDiff finds them much more efficiently.
>
> We also appreciate the references provided and will broaden the related-work discussion accordingly. Our main distinction lies in the scope and operating regime. These methods target asymptotic or particle-based conditional sampling under assumptions such as accurate score estimation, many particles/samples, or repeated refinement. Our paper instead focuses on *practical inference-time control for discrete masked diffusion under hard, black-box, and often non-convex constraints*, including simulator-style evaluators, where exactness is less central than whether one can efficiently generate feasible samples in practice. We will add this discussion in the related work section.
>
> **On rejection sampling.**
>
> We agree that rejection sampling from the pretrained model is conceptually valid. However, in our settings it is often impractical since desirable samples can lie in extremely low-probability regions under the base model. For example, in Fig. 3, we want to generate molecules with QED higher than the maximum QED observed in the training set:
>
> |Method|OOD samples found|Avg. time per OOD sample|
> |-|-|-|
> |MDM + rejection|4.4 ± 0.8 per 32768 samples|4989.67s|
> |SearchDiff|332.4 ± 9.9 per 1024 samples|32.90s|
>
> MDM finds about **1 OOD sample per 7447 generations** on average. This explains why rejection sampling is computationally unattractive in this regime, whereas SearchDiff reaches such regions much more efficiently.
>
> **On the wording about diffusion objectives.**
>
> We agree with your minor correction: discrete diffusion models are trained by maximizing a likelihood-based objective, more precisely an ELBO / masked cross-entropy surrogate, rather than directly "maximizing fidelity." We will revise our wording.
>
> ### Q1. Does the proposed method guarantee sampling within the constrained set? If not, when and why does it fail?
>
> It does not provide such a guarantee. In practice, failure can occur when the denoiser proposal places too little mass near feasible regions or when single-step local refinements are insufficient to cross a large feasibility barrier. This is precisely why success rates differ across tasks: SearchDiff works best when the denoiser proposal and local search together can reach the basin of feasible solutions. Even in those harder settings, however, it substantially improves feasibility relative to the baselines!
>
> ### Q2. Was the evaluation done fairly? Did baselines also use the same evaluation function? Did they use similar computation?
>
> Yes, the comparisons were performed using the same **task-level evaluator** and the same downstream success criteria. However, not all baselines can consume the exact same scalar violation function internally, because the methods are algorithmically different: for example, guidance-based methods use their own guidance mechanisms, while SearchDiff is specifically designed to operate directly on a black-box violation function. We will clarify this distinction in the revision.
>
> On computation, we agree runtime (shown in Tables 6 and 9) is the fairest practical comparison, since different parts of the pipeline run on GPU or CPU. SearchDiff is slower than unconstrained MDM, as expected, but this is justified by the large gains in feasibility, and in sparse feasible regimes, it is dramatically more efficient than rejection sampling, as shown above. We will expand the discussion of this tradeoff and add a clearer limitation paragraph on exactness, evaluator dependence, and runtime scaling.
>
>
> We hope the revised framing of Eq. (3), the clarification of the comparison protocol, and the new rejection-sampling and runtime evidence address your main concerns and convey that the paper makes a practically meaningful contribution in precisely the hard-constrained and OOD regimes where standard alternatives become ineffective.

---

> > ### Author Rebuttal · Reviewer_5gFu · 2026-04-03
> >
> > Thank you for your responses.
> >
> > > It does not guarantee exact sampling from the constrained target. [...] we also aim to reach "out-of-distribution" but feasible regions that have extremely low probability under the base model.
> >
> > I appreciate the clarification, but this raises another concern about the method. If we want to sample OOD (w.r.t. the generative models), what is the point of using the pretrained generative model? Can we just start from a random sample from the training dataset, and then use heuristic search methods (e.g., Large Neighborhood Search, genetic algorithms, etc.) or Bayesian optimization methods combined with the constraint violation functions $\nu$? Related to this point, I believe the experimental results do not provide evidence of good *generative fidelity*, even though the authors consider it a desired feature (line 131 right).
> >
> > Given that the concern is not fully resolved, I will keep my score.

---

> > > ### Author Response · Authors · 2026-04-04
> > >
> > > Thank you for the follow-up. We are glad that the other points have now been clarified.
> > >
> > > On the question of why the pretrained generative model is useful, we first emphasize that the combinatorial spaces considered in this paper are _exceptionally_ large. The pretrained denoiser provides a structured proposal prior at each reverse denoising step, and the search procedure refines these proposals while remaining aligned with the learned reverse dynamics. Without this prior, one is effectively considering a different class of methods, such as seed-and-search approaches or generic unconstrained combinatorial optimization.
> > >
> > > Importantly, our ablations show that the gains do not arise from search alone. As reported in our response to Q2 for reviewer _aQBp_, random initialization followed by Hamming-1 local search, without the MDM denoiser, on the same symbolic tasks yields:
> > >
> > > | Method | Sudoku Acc. (%) | Boolean SAT Acc. (%) |
> > > |:--|--:|--:|
> > > | Random Init + H1-LS | 0.0 | 45.8 |
> > > | SearchDiff | 96.5 | 76.0 |
> > >
> > > These results show that the learned MDM proposal acts as a strong prior that substantially improves both initialization quality and the subsequent denoising trajectory. The same conclusion holds in molecular generation, where this baseline produces only $2.5 \pm 1.4$ valid molecules out of 1024 samples.
> > >
> > > We hope these points clarify the remaining concerns and that you may reconsider your assessment of our work, as the issue does not reflect a factual inconsistency, a missing or unsupported derivation, or an empirical contradiction in the paper, but rather a difference in methodological perspective regarding the role of the pretrained generative prior. Many thanks!

---

### Official Review · Reviewer_udfW · 2026-03-12

**Soundness:** 3
**Presentation:** 3
**Significance:** 3
**Originality:** 3
**Overall Recommendation:** 4
**Confidence:** 4

**Summary:**

The paper presents a method to enforce constraints while sampling masked diffusion models. The idea is to sample a few starting points $x_t$ at stage $t$ and then search in their vicinities (Hamming) for configurations that satisfy the constraints, and use the best fitting sample to go to the next denoising step. The method was evaluated on several domain involving short sequences (biology) as well as Sudoku problems.

**Compliance With Llm Reviewing Policy:**

Affirmed.

**Key Questions For Authors:**

Why only use the best sample from CSS instead of searching around a few of them? How diverse are those samples?

Also, you could train models to work with your search. E.g. https://papers.nips.cc/paper_files/paper/2016/hash/20c9f5700da1088260df60fcc5df2b53-Abstract.html do this for VAE where the encoder's distribution is used to search around for the best posterior for reconstruction; you could be doing something similar in training (although maybe this is follow-up work?)

**Limitations:**

Yes

**Strengths And Weaknesses:**

The idea is sound and the results encouraging.

Equation 3 is suspect. I think I know what the authors had in mind, but I do not think this is mathematically it. In general, $p^C$ cannot be both proportional to $p$ and satisfy the constraint C. The authors might had $p^C(x) \propto p(x)C(x)$ (or $V(x)$ in notation later).

This leads me to the other issue. Running LS R rounds (where R was never specified in the paper but it seems like it was mostly till convergence) can lead the samples very far from the distribution $p(x_t)$, so the sample is not even $\propto p(x)C(x)$. (CSS can be argued to do that, though). On the other hand $p(x_t)$ is trained on data that are close to satisfying the constraint, so this may not be a big issue in practice. Still, given the paper's (a bit overbearing) mathematics, the framing should address this.

---

> ### Author Rebuttal · Authors · 2026-03-30
>
> We thank the reviewer for the thoughtful feedback and, in particular, for noting that the idea is sound and the empirical results are encouraging. We also agree with the points raised about stating Eq. (3) more clearly to distinguish the ideal constrained target distribution from the practical SearchDiff inference procedure. We will revise the paper accordingly.
>
> > **1. Concern with equation 3.**
>
> Thank you for pointing this out. We agree that Eq. (3) should be expanded. What we intended is exactly the constrained conditional distribution suggested in your comment:
> $
> p_{\theta}^{\mathcal C}(x \mid s)
> \propto p_{\theta}(x \mid s)\mathbb{I}[x \in \mathcal C],
> $
> or equivalently,
> $
> p_{\theta}^{\mathcal C}(x \mid s)
> \propto p_{\theta}(x \mid s)\mathbb{I}[V(x)=0].
> $
>
> We will revise Eq. (3) accordingly. More importantly, we will also clarify in the main text that this equation defines the **ideal constrained target**, whereas SearchDiff itself is a **training-free approximate inference procedure** designed to move samples toward that target, not necessarily an exact sampler from it.
>
> > **2. Running LS for $R$ rounds... can move samples very far from the distribution $p(\mathbf{x})$. (CSS can be argued to address that, though). On the other hand, $p(x_t)$ is trained on data that are close to satisfying the constraint, so this may not be a big issue in practice.**
>
> We also agree with this point. Repeated local refinement can move a sample away from the base model distribution, and we do **not** claim that the final output is an exact draw from $p_{\theta}(x \mid x \in \mathcal C)$. Rather, SearchDiff is a practical inference-time method to achieve this goal.
>
> It is worth noting that, for simpler settings such as smooth or convex constraints, one could, in principle, study this effect more formally and derive drift-style guarantees relative to the base proposal (e.g., [Cardei:NeurIPS-25]). However, that is not the setting we target in this paper. Our focus is deliberately on practical and difficult constraint regimes, including non-convex objectives, symbolic constraints, and black-box simulator-based evaluators, where such guarantees are substantially harder to obtain and where inference-time practicality is the primary concern. We will make sure this scope is made explicit in the final version.
>
> As for LS budget, in our main experiments, the default configuration uses **one local refinement per denoising step**. Thus, in the standard setting, search remains closely aligned to the denoiser proposal at every timestep rather than running an unconstrained local optimizer to convergence. To make this concrete, we ran an additional ablation in the same molecular setting as Table 2 with 1-3 LS per denoising step, with results reported in Q3 of our rebuttal to ALkB.
>
> These results show that increasing the number of local refinements beyond the default of 1 yields only marginal gains. We will revise the manuscript to make this design choice explicit and to better distinguish the **conceptual target distribution** from the **heuristic search procedure** used in practice.
>
> ## Questions
>
> > **Q1: Why only use the best sample from CSS instead of searching around a few of them? How diverse are those samples?**
>
> In principle, one could continue search from multiple CSS candidates. Our choice to refine only the best-of-$M$ candidate is primarily a computational one: branching the search around several CSS seeds at every denoising step would multiply the black-box evaluation cost and quickly become expensive. We therefore chose the simplest training-free design that still leverages multiple proposal samples at each step.
>
> Notably, this does not make generation deterministic or collapse diversity. The reverse diffusion process remains stochastic, and CSS itself samples multiple candidates from the denoiser proposal at each step. While we did not include a dedicated within-CSS diversity metric in the submission, we computed pairwise Tanimoto similarity over the final generated molecules as a direct diversity diagnostic reported in Q1 of our response to reviewer ALkB.
>
> The results show the expected tradeoff: SearchDiff is more controlled than unconstrained generation, but it still retains substantial diversity and remains more diverse than stronger constrained baselines such as CBG and TreeG-SC. We will add this analysis to the revision. A multi-branch version that refines several CSS candidates is also a natural extension, and we agree it is an interesting direction for future work.
>
> > **Q2: Also, you could train models to work with your search. [...]**
>
> We thank the reviewer for this interesting connection. We agree that integrating search into training is a promising direction and could potentially yield further improvements. However, this would shift the contribution toward task-specific adaptation and increased training cost. This is a compelling follow-up direction, and we will mention it explicitly in the revision.

---

> > ### Author Rebuttal · Reviewer_udfW · 2026-04-07
> >
> > The authors answered my questions. (The answers were expected; I just wanted to clarify)

---

### Official Review · Reviewer_ALkB · 2026-03-12

**Soundness:** 3
**Presentation:** 3
**Significance:** 3
**Originality:** 3
**Overall Recommendation:** 5
**Confidence:** 4

**Summary:**

This paper introduces SearchDiff, a training-free framework that integrates discrete local search into the reverse denoising process of masked diffusion models to enforce hard constraints at inference time. At each denoising step, the method samples candidate sequences from the denoiser's proposal distribution (Candidate Search Sampling), refines the best candidate over single-token edits, and feeds the result into a modified reverse kernel. The approach is evaluated on molecular generation (QM9), peptide design, tRNA design, Sudoku, and Boolean SAT, showing large gains in constraint satisfaction over baselines.

**Compliance With Llm Reviewing Policy:**

Affirmed.

**Key Questions For Authors:**

Can you report pairwise similarity metrics (e.g., Tanimoto similarity for molecules, edit distances for sequences) within generated batches, and compare against baselines? This would clarify whether the strong constraint satisfaction comes at the cost of diversity.


The QED standard deviation of 0.00 across runs: is this a rounding artifact, or is the method genuinely producing near-identical QED distributions every run? What's the actual unrounded variance?


How many local search refinement rounds are typically taken before convergence at each denoising step? This would help readers understand the practical computational cost versus the worst case.

**Limitations:**

Yes.

**Strengths And Weaknesses:**

Strengths
The core idea here is clean and practical. Masked diffusion naturally exposes full sequences at each denoising step, and the authors exploit this to run constraint-aware search over complete candidates rather than partial prefixes. This is a meaningful advantage over autoregressive approaches where you're stuck committing to tokens left-to-right and can't easily repair early mistakes.
The training-free aspect is genuinely useful. You train your diffusion model once on data, then swap in whatever constraint function you want at inference time. For scientific applications where constraints change across tasks, this flexibility matters.
The empirical results are strong. Perfect constraint satisfaction on peptides, a 27x improvement on tRNA joint feasibility, and the jump from 9.6% to 76% on Boolean SAT are all convincing. The ablations in Table 3 are well done and clearly show that local search is the main driver rather than just sampling more candidates. The out-of-distribution discovery results (Figure 3) are a nice addition that demonstrates the method finds solutions the base model wouldn't reach on its own.
The formalization is clear and the pseudocode in Algorithm 1 is easy to follow. The paper is well written overall.
Weaknesses
The main gap is diversity analysis. The paper reports uniqueness and novelty, but these are weak proxies. There are no pairwise similarity metrics (e.g., Tanimoto similarity for molecules) between generated samples. Greedy hill climbing is well known to converge to local optima, and the near-zero variance in some results (QED 0.77 ± 0.00) raises questions about whether the search is collapsing outputs to a narrow region. This isn't necessarily a problem, but it needs to be characterized. Several other baselines also show ± 0.00 on QED, so it may be a reporting precision issue, but the authors should address this directly.
Scalability is worth discussing, though I don't think it's a dealbreaker for the current contributions. The local search evaluates the full Hamming-1 neighborhood at each refinement round, which scales as the number of masked positions times the vocabulary size, with a constraint evaluation for each neighbor. For the domains tested here this is fine, but the paper should acknowledge what happens for longer sequences or more expensive constraint functions. The tRNA timing results already show the cost of expensive constraint evaluations (7.72s vs 1.11s for peptides).

---

> ### Author Rebuttal · Authors · 2026-03-30
>
> Thank you for the thoughtful and positive feedback. We especially appreciate your recognition of the practical value of the training-free design, the strong empirical performance across multiple domains, and the usefulness of the ablations. We also agree with your suggestions regarding diversity and practical search cost. To address these points, we ran additional analyses during the rebuttal period and report them below.
>
> ### Q1. Can you report pairwise similarity metrics within generated batches, and compare against baselines?
>
> This is a great point, and we agree that similarity-based metrics are informative for assessing diversity. We therefore report pairwise diversity across all domains, using Tanimoto similarity (lower is better) for molecules and edit distance (higher is better) for peptides and tRNA:
>
> | Model     | Molecules ↓     | Peptides ↑      | tRNA ↑          |
> |-----------|-----------------|-----------------|-----------------|
> | Standard  | 0.0717 ± 0.0585 | 0.8582 ± 0.0924 | 0.4517 ± 0.0971 |
> | CBG       | 0.1293 ± 0.0684 | 0.8584 ± 0.0908 | 0.4966 ± 0.0462 |
> | TreeG-SC  | 0.1465 ± 0.2017 | 0.7308 ± 0.1149 | 0.3736 ± 0.1508 |
> | SearchDiff| 0.1186 ± 0.0733 | 0.8115 ± 0.0608 | 0.5242 ± 0.0373 |
>
> SearchDiff maintains competitive diversity across all domains (while achieving substantially stronger constrained generation performance). In particular, it remains more diverse than stronger constrained baselines such as CBG (molecules + tRNA) and TreeG-SC, indicating that the gains in feasibility do not come at the cost of severe mode collapse. We will include this analysis in the revised version.
>
> ### Q2. The QED standard deviation of 0.00 across runs raises questions about possible collapse. Is this a rounding artifact? What is the actual unrounded variance?
>
> Yes, this is a rounding artifact. The values in Tables 1 to 3 report the **mean and standard deviation of the run-level mean QED over 5 runs**, following the evaluation protocol used in prior discrete-diffusion work such as UDLM, where similarly small values also round to 0.00 or 0.01. Thus, the entry 0.77 ± 0.00 does **not** mean that all generated molecules have identical QED or that the method collapses to a single narrow mode. Rather, it means that the **average mean QED across runs** is highly stable.
>
> To make this explicit, we computed the unrounded standard deviation of the run-level mean QED for SearchDiff, which is 0.004855293615777466.
>
> We will report this in the revision and increase the precision in the molecular tables so that this is clearer. We note that the Tanimoto analysis above also supports the absence of severe collapse: SearchDiff retains substantial within-batch diversity even while improving constraint satisfaction.
>
> ### Q3. How many local-search refinement rounds are typically taken before convergence at each denoising step?
>
> In the main experiments, our default configuration uses **one local refinement per denoising step**. Conceptually, each candidate generation can be viewed as a global search step; this global-local procedure is repeated sequentially at every denoising step until the last one. We will emphasize this more explicitly in our manuscript.
> To evaluate whether deeper local refinement materially changes the results, we ran the following additional ablation in the same molecular setting as Table 2.
>
> | Num. LS per denoising step | Admissible Molecules | SA ≤ 4           | Mean QED        |
> |----------------------------|----------------------|------------------|-----------------|
> | 1                          | 474.2 ± 8.7          | 470.2 ± 7.8      | 0.73 ± 0.00     |
> | 2                          | 480.0 ± 8.6          | 476.8 ± 7.2      | 0.73 ± 0.00     |
> | 3                          | 477.0 ± 8.4          | 473.6 ± 6.9      | 0.74 ± 0.00     |
>
> Note that the gains beyond one local search are present, but marginal, which suggests that the setting adopted already captures most of the benefit while keeping test-time cost low. We will make this design choice explicit in the revision and add a short discussion of the cost-performance tradeoff.
>
> More broadly, we agree with your scalability comment and will expand the limitations section to note that the cost grows with vocabulary size, sequence length, and evaluator cost, making SearchDiff especially well suited to scientific discrete domains where the vocabulary is moderate and the value of constraint-aware generation justifies additional inference-time computation.
>
>
> We thank the reviewer again for the constructive feedback. We believe these clarifications strengthen the paper's empirical claims even further.
>
> **Reference**
> [1] Schiff, Yair, et al. 2025. "Simple Guidance Mechanisms for Discrete Diffusion Models." arXiv preprint arXiv:2412.10193.

---

> > ### Author Rebuttal · Reviewer_ALkB · 2026-04-05
> >
> > I appreciate the follow-up. I think this is strong work.

---

> > > ### Author Response · Authors · 2026-04-06
> > >
> > > Thank you for your encouraging feedback. We are pleased that our responses have resolved your concerns and appreciate your recognition of the strength of our work.

---

### Official Review · Reviewer_aQBp · 2026-03-13

**Soundness:** 3
**Presentation:** 3
**Significance:** 3
**Originality:** 2
**Overall Recommendation:** 4
**Confidence:** 5

**Summary:**

In this work, the authors develop SearchDiff, which.proposes a training-free, inference-time neurosymbolic augmentation of masked discrete diffusion models (MDMs) that embeds a two-stage search operator (or Candidate Search Sampling (CSS) followed by greedy Hamming-1 local search (LS)) into each reverse denoising step. At each timestep t, the denoiser produces a clean-state proposal, from which M candidates are sampled and ranked by a black-box constraint-violation function; they then iteratively refine the best candidate via single-token edits over a Hamming-1 neighborhood before being used to define a modified reverse kernel. Their method is evaluated on five tasks: QM9 molecular generation (SA, QED), antimicrobial peptide design (not just peptide design, that's different), tRNA generation, Sudoku, and Boolean 3-SAT (n=7 variables, 45 clauses). Results show large gains in constraint satisfaction over MDLM, UDLM, CDD, and TreeG-SC baselines, with particular strength in the tRNA (27x gain on joint feasibility) and Boolean SAT (9.6% -> 76.0%) settings.

**Compliance With Llm Reviewing Policy:**

Affirmed.

**Final Justification:**

Overall, the authors did a great job responding to my concerns, and that alone justifies an increase in score. I would be supportive of acceptance if the other reviewers feel the same.

**Key Questions For Authors:**

1. PepTune is a masked discrete diffusion model for multi-objective peptide generation that operates in the same domain as Section 6.1.2. It is absent from Table 4. The authors should provide a direct comparison.

2. Table 10 shows SearchDiff achieves 71.5% Boolean SAT accuracy at T=1, only marginally below 76.0% at T=20. What is the accuracy of *random uniform initialization + greedy Hamming-1 LS* (no MDM denoiser) on the same 3-SAT instances? If this denoiser-free baseline performs comparably, I just don't think the MDM backbone is contributing meaningfully to SAT accuracy. If the authors can show a strong gap between the two, that would conversely be a compelling positive result for the paper!

3. The per-step vs. last-step ablation in Table 10 covers Sudoku and SAT but not tRNA. For tRNA, does applying search only at the last denoising step collapse constraint C3 (D/anticodon/ variable loop) to near-zero? The authors should report this single result to complete Table 10 and directly support or challenge the claim that per-step integration is most critical for globally coupled structural constraints.

4. The violation function V(x) in Eq. (7) uses lambda_k = 1 by default, but Appendix C.1.1 describes a hierarchical tie-breaking strategy (SA prioritized over QED) not reflected in Eq. (7). What is SA <= 4 satisfaction and mean QED at equal weighting (lambda_SA = lambda_QED = 1, no hierarchy) vs. the current strategy, for the 32 CSS + LS (all steps) row of Table 3? This is defintely the minimum ablation needed to validate Eq. (7) as written.

5. I would like to see how SearchDiff relate to path-planning methods (i.e. P2)? These methods operate in the same inference-time, training-free regime and perform structured search over denoising trajectories. I'd appreciate a direct comparison or a discussion of the algorithmic differences.

**Limitations:**

Yes.

**Strengths And Weaknesses:**

## Strengths

**Soundness.** I believe the method is technically sound. The modified reverse kernel in Eq. (13) is a clean replacement of the standard masked diffusion update, and Algorithm 1 is reproducible. The ablation in Table 3 does a good job of isolating CSS vs. LS contributions, and Figure 3 provides a nice visualization of OOD discovery driven specifically by LS (212.6 ± 11.7 vs. 0.0 for MDLM and CBG).

**Significance.** The training-free, gradient-free design is valuable: it requires no fine-tuning per constraint and is compatible with any pre-trained MDM backbone. The test-time scaling behavior (Figures 5, 8) is well-documented: the No-LS variant scales from mean QED 0.58 at 8 CSS candidates to 0.65 at 128, while the LS variant improves from 0.70 to 0.75. This helps me connect inference compute budget to generation quality, which is appreicated.

**Presentation.** Figure 2 clearly illustrates the three-stage denoising step. The five-task evaluation spans meaningfully different constraint regimes (soft objectives, structural hard constraints, combinatorial satisfiability).

## Weaknesses

1. There are alternative methods to enable "constrained" generation. For example, PepTune (Tang et al., ICML 2025) guides generation from a pre-trained masked discrete diffusion backbone using external black-box property scorers at inference time without any retraining or gradient access. Since this is generally the operational setting of SearchDiff, I'm quite surprised multi-objective optimization of discrete diffusion methods aren't cited and directly compared against. The related work section (Section 2) characterizes the field as consisting of gradient-based guidance (Gruver et al. 2023, Schiff et al. 2024) and tree search (TreeG-SC), which is an incomplete picture at the time of submission. Also, there are even RL-based fine-tuning methods with objectives (like TR2-D2, from the PepTune team) for MDMs as well. Those should be discussed.

2. The constrained target distribution in Eq. (3) formally requires v(x) = 0, describing a feasibility problem. Yet QED maximization, which are the primary metrics in Tables 1 and 3, is a *continuous* objective, not a feasibility constraint. SA <= 4 is a genuine hard constraint while QED is an objective, but both are folded into V(x) via Eq. (7). The hierarchical weighting that prioritizes SA over QED as a tie-breaker (a pretty substantive algorithmic choice) is disclosed only in Appendix C.1.1, never introduced in the main text, and isn't ablated. Table 8 (SA <= 3, QED >= 0.6) then uses QED as a binary hard constraint with not much discussion of how this differs from the continuous maximization setting of Tables 1-3.

3. The paper's central claim that integrating search throughout the trajectory is a key design principle is strongly supported for SAT but nearly irrelevant for molecules, and the authors never discuss this odd asymmetry. Also, the per-step vs. last-step comparison is also entirely absent for tRNA, the task with the paper's largest absolute gain (27x), which makes Table 10 incomplete.

4. SearchDiff at T=1 achieves 71.5% SAT accuracy; at T=20 it reaches 76.0%. The gap attributable to the search operator (vanilla MDM at 9.6% vs. SearchDiff at T=1 at 71.5%) is approximately 62 percentage points, while additional denoising steps contribute only 4.5 points. The authors never report the accuracy of random initialization + greedy Hamming-1 LS without any MDM denoiser. Without this baseline, it's hard for me to compare the contribution of the MDM backbone's learned proposal distribution to SAT accuracy.

5. There are established MDM path-planning methods that perform structured search over discrete generation trajectories at inference time without modifying base model, placing them in the same regime as SearchDiff. But they are absent from both the related work and the experimental comparison. Overall, I recommend the authors broaden their literature review of the MDM space, which is one of the most glaring weaknesses of this work.

---

> ### Author Rebuttal · Authors · 2026-03-30
>
> Thank you for the careful and constructive review. We particularly appreciate your recognition that the proposed method is sound and significant. We agree that the related-work discussion should be broader and relevant work such as PepTune and TR2-D2 should be discussed explicitly.
>
> Additionally, Eq. (3) should be read as an ideal constrained target, while SearchDiff is a training-free approximate inference procedure rather than an exact sampler from that target. In Tables 1 to 3, SA ≤ 4 is treated as the hard feasibility condition, while QED is a soft objective optimized within the feasible set. The search operator uses a lexicographic ranking: infeasible candidates are always ranked below feasible ones, and feasible candidates are then ranked by QED. By contrast, Table 8 studies a joint hard-constraint setting in which QED is thresholded. We agree that this distinction is substantive and will be moved into the main text.
>
>
> ### Q1. PepTune is absent. The authors should provide a direct comparison.
>
> We agree that PepTune is relevant and should be discussed. In the revised manuscript, we will directly compare against MCTG, the inference-time search component presented in PepTune.
>
> |Method|Feasible Samples ↑|Avg QED ↑|# OOD Samples ↑|
> |-|-|-|-|
> |MCTG|233.6 ± 5.4|0.62 ± 0.00|69.2 ± 1.79|
> |SearchDiff|**560.6 ± 14.3**|**0.77 ± 0.00**|**332.4 ± 9.9**|
>
> For a fair comparison, we adapted MCTG to our setting with minimal changes. As MCTG is designed for Pareto-front discovery in a multi-objective setting, whereas our evaluation returns a single final candidate per run, we select the best candidate produced by each MCTG run under the same evaluation criterion used for SearchDiff. These results show that SearchDiff produces substantially more feasible samples (SA ≤ 4, QED ≥ 0.6), achieves higher average QED and discovers more OOD samples for the QED maximization task, while MCTG requires roughly 3× longer inference time in our implementation.
>
> ### Q2. What is the accuracy of random uniform initialization + greedy Hamming-1 LS, with no MDM denoiser?
>
> We evaluated random initialization followed by pure greedy Hamming-1 local search, with no MDM denoiser, on the same symbolic tasks:
>
> |Method|Sudoku Acc. (%)|Boolean SAT Acc. (%)|
> |-|-|-|
> |Random Init + H1-LS|0.0|45.8|
> |SearchDiff|96.5|76.0|
>
> These results show that the gains are not explained by local search alone. The learned MDM proposal provides a strong prior that substantially improves both the initialization quality and the denoising trajectory. For molecular generation, this baseline yields only 2.5 ± 1.4 valid molecules out of 1024 samples.
>
> ### Q3. The per-step vs. last-step comparison is absent for tRNA.
>
> For tRNA, restricting search to the final denoising step substantially degrades performance:
>
> |Approach|All constraints|Novel & Unique|C1|C2|C3|
> |-|-|-|-|-|-|
> |SearchDiff|719.0 ± 105.1|966.6 ± 46.2|964.6 ± 45.8|955.8 ± 51.4|726.0 ± 102.3|
> |Last-Step|128.8 ± 5.1|990.0 ± 2.4|821.8 ± 12.9|793.6 ± 14.7|138.0 ± 3.8|
>
> Thus, last-step-only search causes a roughly 5.6× drop in all-constraint satisfaction, with a similar drop on C3. More broadly, per-step search has a larger effect for SAT and tRNA as their constraints are more globally coupled and harder to repair late. Molecular QED is smoother, but Table 3 still shows that all-step search outperforms last-step-only search and that extra CSS does not replace local search.
>
> ### Q4. What happens under equal weighting, instead of the current hierarchical SA-first strategy?
>
> The current implementation for Tables 1 to 3 is lexicographic. We ran the exact ablation you suggest for the 32-CSS + LS, all-steps setting. Under equal weighting, we obtain:
>
> |Ranking rule|SA ≤ 4|Mean QED|
> |-|-|-|
> |SA-first lexicographic|**470.2 ± 7.8**|0.73 ± 0.00|
> |Equal weighting|439.2 ± 11.6|**0.76 ± 0.00**|
>
> SA-first lexicographic rule leads to a higher number of samples with SA ≤ 4 (the primary objective), though its mean QED is lower. So the weight choice depends on the use case, in this case finding molecules with as high QED as possible among those with SA ≤ 4. This confirms that the hierarchical ranking is a substantive implementation choice. We will revise the text accordingly.
>
> ### Q5. How does SearchDiff relate to path-planning methods such as P2?
>
> P2 and related path planning methods edit the **denoising trajectory** by planning which positions to unmask over time, whereas SearchDiff keeps the diffusion schedule fixed and performs black-box constraint-aware search over **full-sequence candidates at each step**. They are complementary: P2 changes the trajectory policy, while SearchDiff changes the candidate values fed back into the reverse chain. We will add this discussion explicitly.
>
> **References.**
> [1] PepTune: De Novo Generation of Therapeutic Peptides with Multi-Objective-Guided Discrete Diffusion
> [2] TR2-D2: Tree Search Guided Trajectory-Aware Fine-Tuning for Discrete Diffusion
> [3] Path Planning for Masked Diffusion Model Sampling

---

> > ### Author Rebuttal · Reviewer_aQBp · 2026-03-31
> >
> > The authors have performed all of the expected experiments, and the results are overall very strong vs. MCTG and they ran the exact ablations, showing that hierarchical ranking is a key implementation criteria. I also thank the authors for clarifying my misunderstandings vs. Path Planning, etc. I will raise my score to a 4.

---

> > > ### Author Response · Authors · 2026-04-03
> > >
> > > Thank you for the thoughtful follow-up and for taking the time to re-evaluate the paper in light of the additional experiments and clarifications.
> > >
> > > We appreciate your recognition of our new ablation experiments to support the role of hierarchical ranking as a key implementation component. We also believe these results further strengthen the comparison against MCTG.
> > >
> > > Thank you again for your careful engagement and for updating your assessment of the paper.

---

### Decision · Program_Chairs · 2026-04-30

**Decision:**

Reject

**Comment:**

This paper proposes SearchDiff, a training-free method that integrates discrete search into the reverse denoising process of masked diffusion models for constraint-aware generation under non-differentiable and global constraints.

The reviewers were generally positive about the paper’s technical soundness, practical relevance, and empirical performance across a range of tasks. The training-free and black-box nature of the method is a clear practical strength, and the empirical results are solid.

After considering the reviews and discussion as a whole, however, I did not find the paper’s core contribution sufficiently differentiated for acceptance. While the method is effective and well executed, the overall contribution appears somewhat limited in distinction relative to closely related recent work.
The rebuttal was helpful in clarifying several points, and I appreciate the authors’ efforts in addressing the reviewers’ questions. However, these clarifications did not materially change my overall assessment, as the main limitation concerns the level of differentiation of the core contribution rather than technical soundness.

Overall, this is a solid and practically useful paper. On balance, however, I believe it falls below the acceptance bar.